An Atlantic wide assessment of marine heatwaves beyond the surface in an eddy-rich ocean model

Tobias Schulzki<sup>1</sup>, Franziska U. Schwarzkopf<sup>1</sup>, and Arne Biastoch<sup>1,2</sup>

<sup>1</sup>GEOMAR Helmholtz Center for Ocean Research, Kiel, Germany

<sup>2</sup>Christian-Albrechts Universität zu Kiel, Kiel, Germany

Correspondence: Tobias Schulzki (tschulzki@geomar.de)

Abstract. Periods of prolonged anomalously high temperatures in the ocean, known as marine heatwaves (MHWs), can have devastating effects on ecosystems. While MHWs are extensively studied in the near-surface ocean, little is known about MHWs at depth. As continuous observations in space and time are very sparse away from the surface, basin wide studies on MHWs at depth have to rely on models. This introduces additional challenges due to the long adjustment timescale of the deep ocean, resulting in a long-term drift following the model's initialisation. This unrealistic model drift dominates the MHW statistics below approximately 100 m when a fixed baseline is used. As a result, MHW studies at depth require a long model spin-up, or have to apply a linear detrended baseline removing temperature trends. Based on a comparison of two model configurations with eddy-permitting and eddy-rich horizontal resolution, we show that the representation of mesoscale dynamics leads to pronounced differences in the characteristics of MHWs, in particular along the boundaries and along pathways of highly variable currents. While a high horizontal resolution of the model is important, MHW statistics can be calculated on a coarser grid, which largely decreases the amount of data that needs to be processed. Our results highlight the importance of horizontal and vertical heat transport within the ocean on sub-surface, but also on near-surface, MHWs. By investigating the vertical coherence of MHWs in an example region, here the Cape Verde archipelago, we show that MHWs are coherent over layers of a few 100 to 1000 m thickness, independent of the baseline used. These ranges are closely related to the vertical structure of the temperature field.

Copyright statement. TEXT

**Introduction** 

Marine heatwaves (MHWs) are defined as prolonged periods of anomalously high temperature in the ocean (Hobday et al., 2016). As they can have devastating impacts on marine ecosystems they became a major focus of research over the last decade (Smale et al., 2019; Smith et al., 2023). Several recent studies have shown that MHWs are also connected to a range of physical impacts. Berthou et al. (2024) for example connected the 2023 MHW in the eastern Atlantic to higher land temperatures and increased precipitation probability in the UK and Radfar et al. (2024) showed that MHWs can influence the development of

hurricanes. Furthermore, the study of Krüger et al. (2023) suggests that anomalously high ocean temperatures in the North Atlantic might reduce atmospheric heatwaves in Europe, even though they did not explicitly study the impact of MHWs.

25 A variety of studies exist analysing the characteristics and changes that analyse the characteristics of MHWs at the surface from in model and observation based datasets, but there is still limited knowledge about MHWs at depth. Recently a number of studies were published that aim to understand the occurrence of MHWs beyond the surface, but mostly with a regional focus and typically considering the upper few 100 meters only (Großelindemann et al., 2022; Behrens et al., 2019; Sun et al., 2023; Amaya et al., 2023a; Zhang et al., 2023; Schaeffer and Roughan, 2017; Elzahaby and Schaeffer, 2019). Fragkopoulou et al. (2023) used a global ocean reanalysis to study characteristics of MHWs at a limited number of depth levels reaching beyond 2000 m. These studies generally agree on MHWs below the mixed layer having very different characteristics from surface MHWs. Therefore, it is not possible to make statements about deep MHWs based on the sea surface temperature. As a consequence, in vast areas of the ocean the characteristics and drivers of MHWs beyond the surface have not been identified. Although the surface heat flux is undoubtedly important for surface MHWs in many regions (e.g. Holbrook et al., 2019), Großelindemann et al. (2022); Behrens et al. (2019); Elzahaby et al. (2021); Gawarkiewicz et al. (2019); Chen et al. (2022) and Wu and He (2024) highlight the importance of ocean currents and mesoscale features for generating (sub-)surface MHWs. Further, Hövel et al. (2022); Goes et al. (2024) demonstrate that changes in ocean advection can modulate interannual to decadal variability of the MHW frequency and result in a potential source of predictability. Vertical heat transports and mixing within the ocean are also considered important to generate MHWs themselves, or to set the vertical extend of surface forced MHWs (Schaeffer and Roughan, 2017; Chen et al., 2022). Vertical velocities in the ocean typically show much stronger spatial variations than the surface heat flux and could thus lead to a decoupling of surface and sub-surface MHWs. The importance of ocean dynamics is Compared to the near surface, ocean dynamics are likely even more important in the deep compared to the near-surface ocean, but a comprehensive analysis of MHWs throughout the entire water column is currently missing. This especially includes MHWs that occur along the seafloor, which provides a unique habitat for various marine species, such as sponges and corals. These ecosystems exist in shallow seas, but also deep ocean areas (Roberts et al., 2006; Maldonado et al., 2017) and may be vulnerable to MHWs (Marzinelli et al., 2015; Short et al., 2015; Wyatt et al., 2023; Wu and He, 2024).

Since direct temperature measurements are rare beyond the typical depth of ARGO floats (1000 m) and in particular close to bathymetric features, basin wide assessments of MHWs at depth must rely on ocean models, which comes with several challenges.

50

MHWs are commonly defined as prolonged periods of anomalously high temperature above a seasonally varying baseline (Hobday et al., 2016). Nevertheless, differences exist in the methodology used to define this baseline and the corresponding threshold that must be exceeded in order to identify a temperature anomaly as a MHW. Most studies use a 30-year long baseline period that can be placed in the beginning or end of the available timeseries, if it is longer than 30-years (e.g. Guo et al., 2022). For shorter timeseries the full available timeseries is typically used (e.g. Fragkopoulou et al., 2023), but also for longer timeseries the full timeseries may be used (e.g. Großelindemann et al., 2022). A strong debate evolved around the question whether the baseline should be fixed for a historic reference period, or evolve with a globally rising temper-

ature (Oliver et al., 2021; Amaya et al., 2023b) (Oliver et al., 2021; Amaya et al., 2023b; Smith et al., 2025). This question is frequently discussed in the context of future projections, but already over the historic period trends in surface temperature strongly change the characteristics of MHWs over time (Chiswell, 2022). The debate focuses on the interpretation of the results and in general all these approaches yield meaningful results. However, the choice of the baseline becomes even more important for models that often do not just simulate real trends that are tied to the surface forcing or changes in circulation, but also low frequency adjustments to from the initial conditions known as 'model drift' (e.g. Tsujino et al., 2020). Such a model drift only occurs in the model and has no real world counterpart. Since the near-surface ocean typically adjusts much faster than the deep ocean, model drift becomes increasingly important when MHWs are to be studied at mid- and abyssal depths. As a consequence, the impact of model drift has not gained a lot of attention in the surface focused MHW literature, but will be examined here in detail.

Other important questions arise when performing a basin wide assessment of MHWs in models. Hobday et al. (2016) mention that the statistics of MHWs are likely dependent on the temporal and spatial resolution of the dataset. Regarding the temporal resolution, nearly all studies use daily mean records following Hobday et al. (2016). The impact of horizontal resolution is less clear. When interpolating a high-resolution temperature dataset on a coarse grid, local temperature anomalies are reduced, but also variability is reduced. It is not obvious whether these concurring effects cancel out, or if they lead to more/less MHWs that are detected on a coarser grid. As a consequence, it is not clear whether MHWs detected on the native grid of different model and observation based datasets are directly comparable.

In models another layer of complexity is added by the resolution of the model itself. Model resolution strongly changes local temperature variability (e.g. through the presence of eddies), but also large-scale dynamics (more realistic current strength, pathways and variability). This could directly translate into changes of MHW statistics, as eddies were shown to drive MHWs (Großelindemann et al., 2022; Elzahaby and Schaeffer, 2019; Wyatt et al., 2023; Wu and He, 2024) and MHWs at depth occur most frequently along the pathways of deep currents (Fragkopoulou et al., 2023).



Overall, there remains a lack of knowledge regarding MHWs in the deep ocean and critical challenges in detecting them within models. The overarching goal of this study is to provide a manageable dataset suited for comprehensive studies of MHWs and their impacts throughout the entire Atlantic ocean, in particular including the deep ocean. We use a hierarchy of grids to study the impact of dataset and model resolution on the derived MHW statistics. Further, we study the suitability of different baselines to define MHWs at depth, given the added complexity of model drift after initialisation. We aim to provide a detailed evaluation of different MHW detection methodologies when applied to depth levels away from the surface that may be useful for many following studies. At the same time, we investigate the impact of mesoscale dynamics, ocean currents as well as surface forced and model related trends on the occurrence and characteristics of MHWs throughout the entire Atlantic Ocean, including MHWs at the seafloor. In a last step we investigate which processes determine the vertical coherence of MHWs in more detail for a selected region. Here the Cape Verde archipelago in the eastern subtropical Atlantic is chosen as an example, due to its high biological productivity and ecosystems covering a large depth range in a horizontally confined region.

#### 2 Data and Methods

#### 2.1 Model simulations






This study and the resulting dataset of MHWs is based on simulations in VIKING20X (described in detail by Biastoch et al., 2021), an ocean/sea-ice model configuration employing the NEMO code (version 3.6, Madec, 2016) with its two-way nesting capability AGRIF (Debreu et al., 2008), covering. The configuration covers the Atlantic ocean at 1/20° horizontal resolution and 46 vertical z-levels. The bottom-topography is represented by partial steps (Barnier et al., 2006). The two-way nature of the nesting approach not only provides lateral boundary conditions from the hosting global coarse (1/4°) resolution grid (hereafter referred to as host grid) to the high resolution nest, but also frequently updates the former with the solution on the nest grid by interpolation. For tracer variables, such as temperature, the solution on the host grid within the nested area represents a coarsened version of the nest solution. It therefore includes the dynamical impacts of the higher resolution.

Following the OMIP-II protocol (Tsujino et al., 2020) a series of six consecutive hindcast simulations from 1958 to 2019 forced by the JRA55-do atmospheric dataset (Tsujino et al., 2018) have been performed of which the first cycle (referred to as VIKING20X-JRA-OMIP in Biastoch et al., 2021) has been initialized from hydrographic data provided by the World Ocean Atlas 2013 (WOA13; Locarnini et al., 2013; Zweng et al., 2013) and an ocean at rest, while each. Each following cycle has been initialised from the ocean state at the end of the preceding one. While the transition between the cycles is always between 2019 and 1958, each individual cycle has been extended until 2023. From this series, only the first and sixth cycles for the period 1980-2022 are analysed in this study. The first cycle is selected, as it is closest to the observation-based initial state. Especially in the context of high-resolution modelling, often simulations only cover the hindcast period once. At the same time it is subject to a strong model drift that will be investigated later. The 6th cycle represents an equilibrated model state. Because it had the longest spin-up time, model drift is minimised.

VIKING20X was successfully used in various studies, proving the models capability to realistically simulate the large-scale circulation and its variability (Biastoch et al., 2021; Böning et al., 2023; Rühs et al., 2021), as well as the regional circulation in many locations from the surface to the deep ocean (Fox et al., 2022; Schulzki et al., 2024). Furthermore, the model proved highly capable in simulating MHWs on the Northeast U.S. continental shelf (Großelindemann et al., 2022).

A parallel series of simulations has been performed in ORCA025, the hosting configuration of VIKING20X at 1/4° horizontal resolution, following the same strategy (the first two cycles are referred to as ORCA025-JRA-OMIP(-2nd) and described in detail in Biastoch et al., 2021). Here the period 1980-2022 of only the sixth cycle is used. The two hindcast series in VIKING20X and ORCA025 are directly comparable, with the only difference being the increased resolution in the Atlantic Ocean in VIKING20X.

## 2.2 Definition of marine heatwaves

Marine heatwaves are defined based on the definition of Hobday et al. (2016). For each day of the year the climatological mean and 90th percentile of all temperature values within a 11-day window are estimated. Afterwards they are smoothed using

a 31-day moving average. MHWs were detected using the xmhw python package (Petrelli, 2023). A MHW occurs, if the temperature exceeds the seasonally varying 90th percentile for at least 5 days. If the gap between two MHWs is not longer than 2 days, they are considered as a single event.

MHWs are defined locally, meaning the definition is applied <u>separately</u> at each individual grid point of the three dimensional grid without considering information from other grid points. As a result, the definition can be applied to all depth levels without any modification even though temperature variability is typically much weaker in the deep ocean than it is at the surface.

Although most studies apply the Hobday et al. (2016) definition, a variety of different temperature baselines are used to define MHWs. We apply 3 of the most commonly used baselines in this study. Here we follow the terminology introduced by Smith et al. (2025). First, a fixed threshold baseline is used, where the climatology is calculated for the whole time period from all 43 years (1980-2022) analysed here (1980-2022short: fixed 1980-2022). Second, we apply a fixed 30-year baseline that follows the latest period used to define the World Meteorological Organisation (WMO)climate normals (1991-2020; WMO-No.1203, 2017) baseline based on the first 30-years of the timeseries (1980-2009; short: fixed 1991-2020; WMO-No.1203, 2017) baseline baseline. Here temperature anomalies are defined relative to detrended baseline, where the linear trend based on the time period of the temperature (1980-2022. This is achieved by detrending the temperature timeseries at each grid point before calculating the climatology and ) is removed before performing the MHW detection. A non-linear threshold shifting baseline following Chiswell (2022) was also tested, but is only briefly mentioned in the following results. We acknowledge that various other definitions of the baseline were applied in previous studies, linked to different scientific questions and motivations. Nevertheless, our results can often be transferred to other baselines as well, even if their exact definition may slightly differ from the ones used here. For example using a different 30-year baseline period as the one used here does not change the qualitative results of this study.

Annual mean timeseries of frequency, duration and intensity are created by averaging all events that started in a specific year. For the event based characteristics, namely duration and maximum intensity, spatial averages are calculated without taking grid cells into account where no MHWs occur. For the area based characteristics, namely the number of MHWs and number of MHW days that occur in a region, grid cells without MHWs are included in the calculation.

#### 150 2.3 Choice of the horizontal grid





The host grid of VIKING20X represents a coarsened version of the nest temperature that contains mesoscale dynamics (see descripton of the two-way nesting technique above). In order to assess whether the derived MHW statistics are sensitive to the resolution of the temperature dataset, we compare MHWs detected on the VIKING20X nest grid and the coarser host grid. In both cases we only derive statistics in the Atlantic that is covered by both, the nest and the host grids. Here we only show results using a linear detrended baseline, but the conclusions of this section do not change when other baselines are used (not shown).

Statistics for the whole Atlantic do not differ between the coarser host grid and the high-resolution nest grid. This is true for all depth levels, including the ones shown in figure 1a-c.

The horizontal patterns are not dependent on the dataset resolution either (figure 1d,e). Although the grid is coarser as apparent by individual pixels, the patterns are almost the same. The reason for this result is that MHWs do not occur at a single nest grid point. The zonal and meridional de-correlation scales of the daily temperature timeseries (with the mean seasonal cycle removed) are larger than  $0.3^{\circ}$  even in highly variable regions such as the Gulf Stream separation (not shown) and thus larger than the target grid size (1/4°). As a result, interpolation from the 1/20° to the 1/4° grid does not impact the derived MHW statistics. For interpolation onto an even coarser grid differences are expected in certain regions.



Figure 1. Number of MHW days per year in different depths, derived from the nest  $(1/20^{\circ})$  and coarse host  $(1/4^{\circ})$  grids (Atlantic only) of the first and sixth cycles in VIKING20X (V20X-1st and -6th). Maps show the mean (1980-2022) number of MHW days per year at the surface derived on the nest and host grids (6th cycle). The cross indicates the grid point used as an example in figure 2.

To detect MHWs along the coasts, the higher resolution dataset is advantageous due to the more realistic coastline itself. Otherwise, results are still similar along the ocean boundaries (figure 1d,e). The same argument holds for the seafloor, which

is more realistically represented at higher horizontal resolution. Statistics for larger domains can be calculated on the coarser grid, which significantly decreases the computational costs of detecting MHWs. Accordingly, the following analysis, except for the analysis of bottom MHWs, is carried out on the global host grid of VIKING20X only.

## 70 2.4 Heat budget and MHWs in an example region

In order to study the vertical coherence and drivers of MHWs in detail, we calculate the heat budget for focus on an example region, here the Cape Verde archipelago in the eastern subtropical Atlantic –(26°W-22°W, 13.9°N-18.7°N, see figure 8). In order to study the drivers of individual MHW events, well defined start dates of MHWs in the region are necessary. To obtain such start dates for a larger region and not just a single grid point, individual MHW events are defined on each depth level separately based on the spatially averaged temperature within the region.

To link MHWs to specific drivers, we calculate a heat budget for the same region. The heat content of a given depth level OHC(z) [in J] is calculated from:

$$OHC(z) = \rho_0 c_p \int_A \underline{\underline{T}}(\underline{T - T_{ref}}) \triangle z dA \tag{1}$$

Here A is the area of the Cape Verde archipelago (26°W-22°W, 13.9°N-18.7°N, see figure 8) and T the temperature. The temperature is provided in °C, or equivalently Following Lee et al. (2004) and Zhang et al. (2018) we use a reference temperature of 273.15 K time varying, volume integrated reference temperature for each depth level ( $T_{ref}$ ). As a consequence, the horizontal and vertical heat transport reflect external processes rather than internal redistribution of heat.  $\Delta z$  is the grid cell thickness.  $\rho_0 = 1026 \ kg \ m^{-3}$  and  $c_p = 3991.87 \ kg^{-1} \ K^{-1}$  are the reference density and specific heat capacity. The values are taken from the NEMO routine that is used to calculate the surface heat flux.

The surface heat flux itself is stored in the NEMO ocean model output and integrated over the same area A.

The ocean heat transport across all sections bounding the area A within a given depth layer OHT(z) [in W] is calculated from:

$$OHT(z) = \rho_0 c_p \int_L u_\perp \underline{\mathbf{T}}(T - T_{ref}) \triangle z dL \tag{2}$$

Here  $u_{\perp}$  is the velocity perpendicular to the section. L is a 2D section and dL is the length of a section segment.

Similarly, the vertical heat transport  $\frac{Vert(z)}{OHT_w(z)}$  [in W] is calculated from:

$$\underbrace{\underline{Vert}OHT_{w}}(z) = \rho_{0}c_{p} \int_{A} w\underline{\underline{T}(T - T_{ref})} dA \tag{3}$$

where w is the vertical velocity.

As we are interested in anomalies relative to the mean seasonal cycle (see MHW definition), the seasonal cycle is removed for all heat budget terms.

To investigate the heat budget in certain depth ranges, we sum up the terms vertically. The residual between ocean heat content change and net horizontal and vertical heat transport (which includes and the surface heat flux if the upper boundary is the ocean's surface) represents all heat budget terms that are missing from the calculations described above. This includes horizontal diffusion across the lateral boundaries, but more important vertical mixing. To link heat content variability to MHWs, we have calculated anomalies relative to the daily climatology using the same procedure as for MHWs. We apply both, a fixed 30µr (1280-2009) and detrended baseline.

The contribution of different heat budget terms to individual MHW events was estimated by calculating the heat gain associated with each term from 2 days before the MHW onset to its peak day. This heat gain was then divided by the total heat content change over the same period. As a result, the sum of all contributions equals 100%, although individual terms may contribute negatively (i.e., dampen the anomaly) or exceed 100%. Other lead times prior to the event than 2 days were tested. The results are not sensitive to this choice, as long as the integration does not start within a previous heatwave. To ensure a clear separation of events, we have chosen 2 days (see MHW definition).

## 3 Results






## 210 3.1 Impact of long-term trends on MHW statistics

While the choice of the baseline is important for the interpretation of MHWs at the surface from observations, it is even more important in models. Most models do not just contain 'real' trends related to the surface forcing, or changes in circulation, but also trends that arise from an adjustment of the model after initialisation ('model drift'; see for example Tsujino et al., 2020). This model drift can have a different magnitude and sign at different depth levels (figure 2a,b). While in the top 100 m the model adjusts fast and for the period 1980-2022 trends are similar in the 1st and 6th cycle, major differences between the cycles occur at greater depth. Around 1000 m the ocean shows a strong cooling trend in the 1st cycle, while it warms in the 6th cycle. At 2200 m the ocean warms in the 1st cycle, while it slightly cools in the 6th.

The impact of different baseline definitions on the occurrence of MHWs in the presence and absence of model drift is illustrated in figure 2c-h. A location in the deep (2200 m) subtropical North Atlantic is used as an example here (see black cross in figure 1d.e). The temperature in the 1st cycle shows pronounced multi-decadal variability with high temperatures during the first and last 10 years of the timeseries and lower temperatures in between. In the presence of strong model drift (1st cycle) multidecadal variability is only captured by the MHW statistics when a linear detrended baseline is used. With all other the two fixed baselines a linear trend in the temperature leads to MHWs only occurring causes MHWs to occur predominantly in the later period. This is different in the 6th cycle, where, without a linear temperature trendin the absence of model drift, the different baselines yield more similar results. Also, for the linear detrended baseline the results are more similar between the two cycles. While the correlation between the timeseries of the 1st and 6th cycle is higher than 0.9 at the surface and in 100 m for all baselines, there is a weak anti-correlation (about -0.3) in 1000 m and 2200 m for the WMO and fixed baselines. The linear baseline timeseries are still highly correlated between the 1st and 6th cycles in 1000 m (0.91) and in 2200 m (0.85).

On the one hand this, suggests that the MHWs derived at this particular location in the 1st cycle are dominated by unrealistic drift, except when a linear detrended baseline is used. On the other hand, it suggests the linear temperature trend over the last 40-years was of minor importance . Instead in the deep (2200 m) subtropical North Atlantic when model drift is absent (6th cycle). Instead, changes in the occurrence and characteristics of MHWs were dominated by (multi-)decadal variability, independent of the baseline used for detection.

The impact of model drift in temperature is directly reflected in the basin wide MHW statistics as well. At the surface and in 100 m depth MHW statistics are robust across the cycles (figure 3a,b upper two rows), suggesting that the trend is mostly caused by the common surface forcing. In contrast, a strong drift in the temperature of the 1st cycle leads to major differences at depth, if the fixed and WMO baselines are applied (figure 3a,b lower two rows). At 1000 m depth, the number of MHW days is small before and increases after 2000 in the 6th cycle, but. Conversely, it shows a sharp decline in the 1980s and then stays near zero in the 1st cycle. At 2200m the ocean reaches a near-permanent MHW state toward the end of the timeseries (365 MHW days per year) in the 1st cycle. The strong increase in MHW days is more pronounced with the WMO fixed 30 yr. compared to the fixed 43 yr. baseline. This is because the threshold at 2200 m is lower when the first last 10 years are not included, since the mid-depth ocean was warmer in the 80s than in the 90s in anomalously warm in the 2010s in most regions (see figure 2 as an example). In contrast, the 6th cycle shows a decrease in MHW days from around 100 to almost no MHW days per year until after the 1990s.

Using a linear baseline (detrending) Relative to a fixed baseline, the evolution of the MHW statistics is dominated by the impact of model drift in the 1st cycle. Using a detrended baseline removes most of the model drift, but it also removes any forcing related trends. This is particularly visible at the surface and in 100 m depth. Therefore, (figure 3a,b upper two rows). As it is not possible to recover know from the experiments themselves which part of the trend is related to the forcing and which part to model drift, there is no method to recover the fixed baseline statistics of the MHW statistics from non-detrended temperatures in the 6th cycle by detrending from the 1st cycle(e.g. 1st-linear does not match 6th-fixed/WMO). As a . As consequence, the 1st cycle can not be used to make any statements about the long-term evolution of MHWs at depth relative to a fixed baselineas the evolution is dominated by the impact of model drift.

Applying the linear baseline in the 1st and 6th cycle detrended baseline yields more similar, but not identical, results (in the presence (1st cycle) and absence (6th cycle) of model drift (red lines in figure 3a,b)—lower two rows). While the correlation between the timeseries of the 1st and 6th cycle is higher than 0.9 at the surface and in 100 m for all baselines, there is a weak anti correlation (about -0.3) in 1000 m and 2200 m for the fixed baselines. The detrended baseline timeseries are still highly correlated between the 1st and 6th cycles in 1000 m (0.91) and in 2200 m (0.85). Detrending also leads to a similar evolution of the temperature itself in both cycles (e.g. figure 2e,f). This is also true for the global mean temperature at all depth levels (not shown). Therefore, assuming Assuming that model drift adds linearly to forced temperature trends is therefore a reasonable assumption, but non-linear adjustments are not completely absent. A non-linear shifting baseline was tested (not shown) as well, but yields no advantage over the linear detrended baseline. In agreement with Chiswell (2022), differences in the derived MHW statistics are smalland. As the non-linear shifting baseline has disadvantages due to the finite length of the timeseries. A

**Figure 2.** Globally averaged temperature anomaly relative to the 1980-2022 mean, illustrating model drift (a,b). Temperature relative to the MHW threshold estimated with different baselines at 41.8°N, 38.6°W (see cross in figure 1d,e) at 2200 m depth (c-h).

moving average leads to a loss of data at the beginning and end of the timeseries, or the averaging window has to be modified at the beginning and towards the end which can introduce spurious signals. Therefore the non-linear shifting baseline is not

Figure 3. Number of MHW days per year averaged over the Atlantic in the 1st, strongly drifting (a) and 6th, equilibrated (b) cycle at selected depths and for different baselines. Depth profiles of the difference in the number of MHW days between the last (2013-22) and first ten years (1980-89) for the entire Atlantic (c) and for the Cape Verde archipelago (d; dashed) and Labrador Sea (d; solid). For the entire Atlantic the linear detrended and WMO fixed 30yr baselines are shown, while in d) only results from the WMO fixed 30yr baseline are shown.

Vertical profiles of differences between the last and first ten years of the timeseries further reveal that only in the top 100 m statistics are comparable between the 1st and 6th cycle when the WMO-fixed<sub>30yr</sub> baseline is used (figure 3c). Below 100 m depth, the number of MHW days increases stronger in the 1st cycle shows a stronger increase from the beginning to the end of the timeseries in the 1st cycle. Below 600 m depth changes are of opposite sign in the 1st and 6th cycle. Thus, model drift dominates the long term changes in MHW statistics of the first cycle below approximately 100 m depth. When using

a detrended baseline, changes in the mean temperature between the decades are mostly removed and the resulting changes in MHW days are almost zero. This indicates that changes in the shape of the temperature distribution (standard deviation, skewness) that could lead to differences between the decades with a detrended baseline played a minor role.

There are regional differences, however (figure 3d). For example in the Labrador Sea (solid) results differ already at the surface. In the eastern subtropical Atlantic (Cape Verde archipelago; dashed), changes are similar within the top 50 m. This highlights that the impact of model drift is not the same everywhere. Note that the surface flux is (almost) the same in the 1st and 6th cycle and thus can not explain differences between the cycles. In regions where ocean advection and stratification play an important role near-surface, model drift is likely to affect the occurrence of MHWs at depths even shallower than 100 m. This is the case for the Labrador Sea, where advective processes strongly influence the mixed layer dynamics (e.g. Gelderloos et al., 2011). For the Cape Verde archipelago variability in the mixed layer, that in the annual mean extends to approximately 50-70 m depth, is strongly influenced by the surface heat flux as will be shown later (section 3.4).

In any case, model drift dominates long term changes in MHW statistics away from the surface. Thus, MHW detection at depths beyond approximately 100 m requires a well spun-up model, if the impact of long term trends (e.g. related to climate change) is to be studied. Model studies that focus on the surface or shallow seas are able to do so using a shorter model spin-up. With a linear baseline, statistics are not identical, but similar in the following we focus on the characteristics of MHWs detected by applying the fixed<sub>3000</sub> baseline in the presence (1st cycle) and absence (6th cycle) of strong model drift, as the occurrence of MHWsis mostly related to interannual to decadal variability. Studies applying a linear baseline are therefore possible with a shorter model-cycle. As argued above, the 1st cycle can only be used to study MHWs defined based on the detrended baseline, but we are explicitly interested in studying the impact of long-term changes in the surface forcing and ocean circulation on MHWs. Because including the trend requires a very long-model spin-up as well. In the following we focus on the that is rare at 1/20° resolution, this provides unique and novel insights into the characteristics of MHWs detected by applying the WMO baseline in the 6th cycleand the impact of long-term temperature changes. Even though the trends of temperature trends in the deep ocean temperature are highly uncertain due to the lack of long-term observations, the well spun-up experiment 6th cycle is regarded as the best estimate available.

#### 3.2 Horizontal and vertical changes in MHW characteristics

#### 3.2.1 Characteristics of MHWs at the surface







Compared to the observation based NOAA OISST dataset (Huang et al., 2021), VIKING20X overestimates the duration, but underestimates the maximum intensity of MHWs at the surface throughout most of the Atlantic (figure 4; WMO fixed<sub>30ur</sub> baseline). This is a well documented feature of many models compared to satellite based datasets (Qiu et al., 2021; Pilo et al., 2019). A notable exception is the northern flank of the GS, where VIKING20X simulates higher maximum intensities. The temporal variability of the basin averaged statistics are in very good agreement, however (figure 4g-i). The correlation exceeds 0.66 for all timeseries. While the magnitude of variability is similar for the frequency and maximum intensity, it is higher for

the MHW duration in the model. The duration and frequency show a positive linear trend, which is slightly stronger in the NOAA OISST dataset. The maximum intensity does not show a clear trend in neither of the two datasets.

The horizontal patterns of the time mean frequency and maximum intensity match the observation based product as well (note the difference in the colorbar that represents the mentioned mean bias). MHW frequency is high A high frequency of MHWs is seen along the equator, in the western North Atlantic subtropical gyre and in a zonal band around 30°S (figure 4b.e). Only few MHWs per year are detected within 20° around the equator and in the eastern subpolar gyre in both, the model and observations. The Gulf Stream (GS), boundary currents of the subpolar gyre and the upwelling regions along the eastern boundary stand out with high maximum intensities (figure 4c.f). Differences between the datasets are more pronounced for the duration of MHWs (figure 4a,d). In particular between the equator and 20° Nthe duration is much shorter in observations. The model and observations agree on regions with longer and shorter duration in most other regions, but the difference between regions of high and low mean durationis more pronounced in the model, VIKING20X shows very long, but OISST very short MHWs. In most other regions the model and satellite data agree on whether MHWs are comparably long or short (e.g. long MHWs North of the GS separation and short around the UK), but VIKING20X generally overestimates the duration. Differences in duration are largest in regions that show high cloud cover, especially in the Intertropical Convergence Zone (ITCZ), limiting the availability of satellite based SST. This may contribute to the larger difference in these regions compared to other parts of the Atlantic. Nevertheless, it can not be concluded that the model is more realistic and model biases (e.g. caused by limited vertical resolution) could play an important role too. For example the vertical resolution of the model can be important in regions that develop very shallow mixed layers during surface forced MHWs. The heat gained through the surface is then mixed over a smaller volume, which may not be adequately simulated by a model with too coarse vertical resolution.

## 3.2.2 Characteristics of MHWs at depth





A broken-section through the Atlantic (see black lines in figure 4d-f) shows that the characteristics of MHWs considerably vary in the horizontal, as well as in the vertical plane (figure 5). MHWs in the The section was chosen to compare the vertical structure of MHWs in dynamically very different regions, starting in the South Atlantic, crossing the equator, the Mid-Atlantic Ridge, the western boundary current system and finally the deep convection region in the subpolar gyre. MHWs in the abyssal ocean last long, but occur rarely. This is Duration and frequency are directly related, as MHWs that last a year can, by definition, only occur once a year. The only exception is the eastern subtropical gyre, between Puerto Rico and Canada, where relatively short MHWs occur at abyssal depth. However, the abyssal ocean (deeper than approximately 4500 m) may not have fully adjusted even after more than 300 model years. Therefore, the abyssal ocean should be interpreted with caution. Another maximum in MHW duration along the section occurs between 1000 and 2000 m. In the subpolar gyrethe depth range extends from, between Canada and Greenland, very long MHWs occur between 300 to and 3000 m depth (figure 5a).

There is a general tendency for longer MHWs to coincide with a lower frequency, MHWs to be longer in regions where the frequency is lower (figure 5a,b), but compared to the duration, the frequency shows stronger differences along the section at the same depth level. The frequency is generally higher along the boundaries. For example in the eastern upwelling regions

**Figure 4.** Mean (1982-2022) duration, frequency and maximum intensity of MHWs at the surface in the NOAA OISST dataset (a-c) and the 6th cycle of VIKING20X (d-f). The black line in d-f indicates the section shown in figure 5 (letters mark the section vertices). Timeseries show the annual mean MHW characteristics averaged over the Atlantic from both datasets (g-i).

(around Africa) and within the western boundary currents . Here (around Puerto Rico), the highest frequency is not reached at the surface, but between 500 and 1000 m depth. In the subpolar gyre the maximum frequency is reached even deeper.

The maximum intensity peaks at the surface along the entire section (figure 5c). Outside the tropics, a secondary maximum occurs at depths around 500 m. Below 1000 m, the maximum intensity does not exceed 0.1°C in most regions. Within energetic currents, for example the Deep Western Boundary Current (DWBC) close to Puerto Rico and the GS, the intensity is higher and can reach up to 0.2°C even beyond 3000 m depth (figure 5c). Also around the Mid-Atlantic Ridge (MAR), the maximum intensity is elevated. In the deep convection region in the western subpolar gyre (between Greenland and Canada) the maximum intensity is higher than 0.1°C down to 2500 m depth.

Overall, the sections show that MHWs are more frequent and intense in regions of strong current variability and/or strong gradients in mean temperature. The presence of deep currents leads to pronounced sub-surface maxima in frequency and intensity, with a tendency for comparably short MHWs. Variability linked to deep convection in the central Labrador Sea (between Greenland and Canada) causes intense MHWs below 1000 m depth as well, but they last longer and occur less frequent than MHWs along the deep boundaries.

**Figure 5.** Mean (1980-2022) duration, frequency and maximum intensity of MHWs along a section through the Atlantic Ocean (see figure 4) based on the 6th cycle of VIKING20X and applying the WMO fixed 30 yr baseline (shading). The sign of the linear trend (1980-2022) at the same grid points is indicated by crosses (positive trend) and dots (negative trend). They are only drawn where the trend is significantly different from zero based on a 5% significance level.

Many studies and the VIKING20X model agree on an increase in MHW frequency, duration and intensity at the surface (Xu et al., 2022; Oliver et al., 2018; Chiswell, 2022). The magnitude and sign of linear trends is not uniform across the ocean, however. The duration of MHWs increased in the tropical and central subtropical gyre Atlantic (around point B) between 500 and 1500 m depth (crosses in figure 5a). The South Atlantic (between points A and B) shows a significant decrease in duration (dots in figure 5) in duration a) below 2000 m, while the tropical and subtropical North Atlantic only shows (B to D) only show such a decrease between 2000 and 3500 m. Regions of positive trends in duration are also subject to a positive trend in frequency and maximum intensity (figure 5b.c). This may seem to contradict the previous description, as MHWs in regions of higher mean frequency tend to be shorter. At the same time it is expected, since a warming leads to the threshold being exceeded more often and for longer. A distinct pattern can be seen in the subpolar gyre (D to E). Here positive trends in all characteristics occur at the surface and negative trends around 3000 m (figure 5a-c), which can be explained by a reduction in deep convection. A near-surface warming increases the MHW intensity, duration and frequency, but also reduces the mixed layer depth. The shallower winter mixed layers then prevent mixing between the deep waters (that are no longer reached by the mixed layer) and the upper ocean waters. Below 3000 m the boundary currents are colder than the interior Labrador Sea, likely causing a cooling in the absence of exchange with the surface.

#### 3.2.3 Bottom marine heatwaves







As the seafloor provides a unique habitat for various marine species, the detection of MHWs along the seafloor rather than a fixed depth is of major interest for the biological community. Bottom MHWs are defined here as MHWs that occur in the last ocean filled model grid cell above the bottom.

The deep ocean basins with depths exceeding 5000 m are characterised by very long, but infrequent MHWs (figure 6a-ea,b). The duration of MHWs exceeds a year and therefore the low frequency is a direct consequence, as already mentioned above. Also, Additional analysis performed on bottom MHWs detected with a detrended baseline (not shown) suggests that the bottom water masses are experiencing a long-term warming trend. The source of the long-term warming trend can not be known, but the bottom water masses are potentially probably still subject to model drift, even after more than 300 years of model spin-up, which due to the long adjustment timescale of the abyssal ocean. This leads to a near permanent heatwave state in the beginning or end of the timeseries. Although, their maximum intensity remains below 0.1°C (figure 6c), the temperature tolerance of most deep sea species is highly uncertain and the possible impact of such low intensity, but long lasting MHWs, yet to be determined.

It is interesting to note that, even though very deep, the North American Basin is characterised by rather short MHWs (figure 6a,b). This suggests that the high variability associated with the GS impacts temperature extremes down to the sea-floor. In general, bottom MHWs are shorter near and along the continental slopes compared to the interior ocean abyssal plains and Mid-Atlantic Ridge, even if the sea-floor is located at similar depths. This is not just related to different depths, as the bottom along the lower continental slope is deeper than parts of the Mid-Atlantic Ridge. The frequency of bottom MHWs is highest along the continental slope (figure 6a,b). Notably it is higher along the slope than it is on the shelf, which is related to the subsurface maxima along the slope seen in figure 5. High frequencies are seen along the western boundary following the DWBC pathway, along the pathways of the overflow water in the subpolar gyre and along the eastern boundary. The maximum

**Figure 6.** Mean (1980-2022) duration, frequency and maximum intensity of MHWs at the bottom (WMO-fixed<sub>30ur</sub> baseline; VIKING20X-6th nest grid; a-c). The linear trend is shown in the lower panels (d-f). Significant trends (5% significance level) are indicated by dots. A bottom depth of 5000 m is indicated by the white contour.

intensity strongly follows the bathymetry with highest intensities reached on the shelf. Further, the intensity is higher along seamount chains and the Mid-Atlantic Ridge where the sea-floor is elevated (figure 6c).

Bottom MHWs on the shelves show an increase in frequency, duration and maximum intensity over time (figure 6d-f). The linear trend in these MHW characteristics is not significant everywhere on the shelf and along the upper continental slope, due to strong interannual variability. Significant positive trends in deeper ocean regions frequency, duration and maximum intensity can be seen in the eastern subtropical North Atlantic and the western tropical Atlantic. Significant negative trends can be seen occur in the subpolar gyre, Caribbean Sea and in the entire eastern tropical and subtropical South Atlantic. Again, trends in the deep ocean are associated with higher uncertainties, due to the lack of long-term observations and in. In particular the abyssal plains may be still subject to model drift.

#### 3.3 The impact of mesoscale dynamics on the characteristics of MHWs



The previous sections suggest that the presence of strong and highly variable currents has an important impact on the characteristics of MHWs. To test how the representation of mesoscale dynamics impacts the characteristics of MHWs, we compare the 6th cycle in VIKING20X to the 6th cycle in the un-nested ORCA025 configuration. The 6th cycle is chosen here, because

we aim to compare MHWs in a adjusted ocean state. The temperature drift (and thus MHW statistics) is different in ORCA025 and VIKING20X throughout the first cycles, such that the results are not directly comparable. The higher horizontal resolution leads to the presence of individual mesoscale features in VIKING20X, of which only a small part is simulated in ORCA025 outside the subtropics (Biastoch et al., 2021). It also leads to changes in the mean current structure and position, as well as differences in temperature trends and in horizontal and vertical temperature gradients (figure A1). All these aspects may have an impact on the characteristics of MHWs.





The VIKING20X dataset contains the imprint of mesoscale dynamics throughout the Atlantic as discussed above, although for both configurations MHWs are detected on a 1/4° grid. Results are only shown for the WMO fixed<sub>30yr</sub> baseline, but the conclusions do not depend on the baseline.

At 1000 m depth VIKING20X shows a high frequency of MHWs along the western and eastern boundaries as well as along the equator (figure 7a). Compared to ORCA025 the frequency in VIKING20X is higher almost everywhere, but in particular along the western boundary and eastern boundary outside the tropics (figure 7c,e). At the western boundary this goes along with a stronger, more narrow and more variable western boundary current in VIKING20X. This is apparent by the difference in mean and eddy kinetic energy (MKE/EKE) in the two configurations(figure A1). Also vertical velocity fluctuations (vertical velocity EKE) are stronger along the western boundary in VIKING20X (figure A1a-c). Along the eastern boundary differences in MKE are relatively small, but EKE is still higher at least north of the equator. Also the horizontal and vertical temperature gradients are larger in VIKING20X (figure A1d,e). When averaged over the entire Atlantic, the temporal evolution of the MHW frequency is similar (correlation of 0.94), but the mean frequency is clearly higher in VIKING20X (figure 7e). Although it is not possible to prove that the higher frequency in VIKING20X is more realistic, it is expected that the more realistic representation of currents and their variability leads to more realistic MHW characteristics in VIKING20X.

The maximum intensity shows a strong maximum in the region of the GS separation at 34°N (figure 7b). Also in the eastern Atlantic between 40°N and 60°N the mean maximum intensity exceeds 1.2°C. In this area, the transition between the warm Mediterranean Sea Outflow Water and colder North Atlantic Deep Water or Subarctic Intermediate Water is located at approximately 1000 m depth (Kaboth-Bahr et al., 2021; Liu and Tanhua, 2021), suggesting that a large. The transition between these water masses of different temperature is related to a strong vertical temperature gradient could be the cause for the high MHW intensity (see Liu and Tanhua, 2021). Therefore, vertical displacements of isotherms, for example through 425 internal waves or other vertical velocity anomalies, can cause strong temperature anomalies in these regions. High values of maximum intensity are further seen west of Greenland, where the flow instabilities lead to the shedding of West Greenland Current eddies and Irminger Rings, which are typical expressions of the mesoscale resolution in this region (Biastoch et al., 2021; Rieck et al., 2019). Along the western boundary, south of the DWBC/GS crossover, the maximum intensity is higher 430 than in the interior as well. Compared to ORCA025, VIKING20X simulates more intense MHWs (figure 7d,f). The difference is strongest in the regions where the mean maximum intensity is highest. West of Greenland this is directly linked to the presence of mesoscale eddies in VIKING20X as discussed above. The presence of Irminger Rings that transport warm water from the Irminger Current into the cold central Labrador Sea in VIKING20X is likely related to the occurrence of strong

Figure 7. Mean (1980-2022) MHW frequency and maximum intensity at 1000 m depth in the 6th VIKING20X cycle (a,b). Difference between VIKING20X (6th cycle) and ORCA025 (6th cycle) (c,d). Timeseries show the average over the Atlantic from both experiments (e,f). In all panels MHWs are defined using the  $\frac{\text{WMO}}{\text{fixed}_{30yr}}$  baseline.

MHWs. The negative difference north and positive difference south of the GS pathway suggests a more southern position of 435 the GS in VIKING20X. This is supported by the MKE and horizontal temperature gradient differences (figure A1a). Higher intensities are seen along most of the western boundary related to a stronger horizontal temperature gradient and higher EKE (figure A1c,e). In the eastern North Atlantic the vertical temperature gradient is stronger in VIKING20X (figure A1d), which can explain the higher maximum intensities. As explained above, the vertical temperature gradient is strongly influenced by the properties of the Mediterranean Sea outflow in this region. The reason for the different stratification may not be directly related to the representation of the mesoscale, but it demonstrates the importance of the outflow properties on the characteristics of

#### MHWs in the eastern mid-latitude North Atlantic.

Overall, the more realistic representation of boundary currents, coastal upwelling and the ability to resolve sharper temperature gradients leads to a higher frequency (and shorter duration) and higher maximum intensity of MHWs throughout most of the Atlantic, but in particular along the western boundary. This was only shown for the depth of 1000 m, but similar arguments apply at least to the depth range from 300 to 3000 m. This means that studying the impact of MHWs on deep ecosystems along the continental slopes requires models with sufficiently high resolution. Away from the boundaries the model resolution is particularly important along the GS and North Atlantic Current (NAC)pathway throughout the entire water column(not shown).

## 450 3.4 Vertical structure and drivers of marine heatwaves in an example region

In order to better understand the vertical structure and coherence of MHW events, we now focus on the Cape Verde archipelago in more detail. The Cape Verde archipelago is located in the eastern tropical Atlantic in between the westward flowing North Equatorial Current and eastward North Equatorial Counter Current. It is part of the Canary Current upwelling system and thus characterised by large-scale upwelling (Arístegui et al., 2009; Cropper et al., 2014).

This region is selected here as an example, due to its high species richness, including the presence of vulnerable marine ecosystem (VME) indicator species (Vinha et al., 2024; Hoving et al., 2020; Stenvers et al., 2021) and large bottom depth gradients (see figure 8) such that benthic ecosystems span a large depth range in a horizontally confined region. As a result, one might expect MHWs to have important impacts beyond the surface and thus a detailed understanding of their characteristics throughout the water column is highly relevant. Furthermore Additionally, the eastern subtropical Atlantic was identified as a region of significant positive is interesting in the context of this study, as it shows significant trends in the MHW characteristics in approximately the top 1000 m and negative trends below that change sign at approximately 1500 m depth (figure 5).

#### 3.4.1 Variability of MHW characteristics

470

The characteristics of MHWs at the surface do not vary in phase with the characteristics of MHWs below approximately and below 50 m depth do not vary coherently (figure 9). a-c). For example a period of short MHWs is the 2010s near-surface coincides with a period of anomalously long MHWs between 100 and 1300 m. Already at 100 m depth the explained variance (squared correlation coefficient  $R^2$ ) by their respective surface values has dropped to values below 25% for all MHW metrics (figure 9d-f). This means that the surface heat flux variability of the surface metrics, which are strongly linked to variability of the surface heatflux as will be shown later, accounts for only a small fraction of the variance in the MHW characteristics at depth. This is consistent with impact of surface forced trends being limited to about 50 m depth in the Cape Verde archipelago as discussed above (figure 3). Removing the Different long-term trends could decouple surface MHWs from deeper MHWs regardless whether the drivers of individual heatwaves act coherently across a larger part of the water column. However, removing the linear trend of the MHW characteristics itself, or detecting MHWs with a linear detrended baseline leads to the

**Figure 8.** Depth of the seafloor in the eastern tropical Atlantic. The Cape Verde archipelago region (red) is used for the spatial averages in figures 9, ?? 10 and ?? 11.

same results . It is therefore not just different long-term trends that decouple surface MHWs from deeper MHWs, but also different characteristics (figure 9d-f). This suggests that the different drivers of variability on shorter timescales rather than different long-term trends are the main reason for the low correlation between the MHWs metrics at the surface and at greater depth.

480

485

Annual mean duration, frequency and maximum intensity are subject to pronounced interannual variability near-surface. In deeper layers low frequency variability is more dominant (figure 9a-c), which leads to a lower explained variance. In the late 2000s an increase in MHW duration, frequency and maximum intensity seems to propagate from the surface towards sub-surface layers above 300 m. Thus, there might be a connection between surface MHWs and MHWs in the upper 300 m even though they don't have the same characteristics, nor appear at the same time. The maximum intensity is scaled with the difference between the climatological mean and 90th percentile here to compare the anomaly to the magnitude of local variability, which is much lower in the deep ocean. It is interesting to note that the absolute maximum intensity is much stronger near-surface (see for example figure 5c), but the scaled intensity shows similar values at all depth levels (figure 9c). This means relative to the typical range of variability, MHWs are similarly intense temperature variability, the anomaly associated with MHWs is similarly strong at all depths.

Between 300 and 1200-1300 m MHW characteristics are mostly in-phase between vary coherently, with low values in all variables before 2000 and higher values afterwards (figure 9a-c). For this depth range no connection to the surface can be identified, even when possible time delays are considered. Between 1200-1300 m and 3000 m strong long and intense MHWs occur frequently at the beginning of the timeseries and nearly no MHWs occur after 1995. Between 3000 and 4000 m MHWs do not show a considerable trend, but more there is a small (compared to other depth layers) decreasing trend in frequency and duration, but variability on interannual timescales dominates the timeseries. It is important to note remember that the MHW definition was applied at individual grid points and depth levels and does not include any spatial information (see methods). Still, the characteristics of MHWs vary in phase over larger depth ranges, even though mostly not connected MHWs characteristics exhibit a similar temporal evolution over broader depth ranges. This is primarily not related to the surface forcing, but other processes that act coherently over a certain depth ranges. These processes are investigated in more detail in the following.

## 3.4.2 Vertical coherence of individual MHWs

In order to understand which drivers control the variations of MHW characteristics at different depth, we now consider individual MHW events and investigate the daily fraction of the total area that is covered by a MHW relative to the entire area of the in the Cape Verde archipelago(figure ??). In general, the annual mean characteristics described before (figure 9a,b) are also reflected by the individual events (figure 10a,b). The mixed layer is characterised by intermittent events where most of the area is in a MHW state. High MHW coverage often only persists for a few months. After 1995 high MHW coverages above 70% persists for longer, sometimes more than a year, relatively short, intermittent events. By comparing the results from MHWs detected using the WMO and linear baselines, fixed<sub>30wr</sub>, and detrended baselines (figure 10a,b), it is evident that the linear trend in within the mixed layer had a minor impact on the occurrence of MHWs in the Cape Verde archipelago. The timing and coverage of MHWs is similar (figure ??a,b)Compared to deeper depth levels, the timing of MHWs events is more similar near the surface. The mixed layer (here maximum mixed layer within the region) shows a seasonal cycle from 20 m in summer to 100 m in winter. In several years (e.g. 1995, 1998, 20022005, 2010) MHWs occur in the mixed layer when it is deep during winter and remain below the mixed layer as it shoals in summer (figure 10a,b). While MHWs are then often terminated in the mixed layer, presumably due to surface fluxes, they persist below. This leads to different characteristics of MHWs in the top 50 m and the depth layer between 50 and 100 m, but MHWs may still be initially forced by a heat gain through the surface.

Below 100 m only in few years (e.g. 1992, depth some MHWs are connected to surface MHWs (e.g. 1995, 2005, 2010)a high MHW coverage at the surface seems to propagate to 300 m depth in the following years. However, in particular when using the fixed<sub>30yr</sub> baseline (figure 10a,b). In other years MHWs occur without any apparent link to the surface (e.g. 1990 for both baselines and 2014, 2020 with the detrended baseline). Therefore, the depth range between 100 and 300 m is elearly not only governed by the surface exchange of heat, but other processes as well. The MHW coverage shows heat content and occurrence of MHWs show a stronger positive trend in this depth range compared to the surface (figure 10a). MHWs detected based on the linear baseline also show an increase after detrended baseline also occur more often between 2004, but the coverage is generally lower and 2020, but they are generally shorter (figure 10b). This indicates that both, decadal variability and a long-

Figure 9. Annual mean MHW duration, frequency and scaled maximum intensity in the Cape Verde archipelago (WMO-fixed<sub>30yr</sub> baseline; VIKING20X-6th; a-c). Explained variance of the MHW duration, frequency and scaled maximum intensity at different depth by their annual mean values at the surface (d-f). The explained variance was calculated for the WMO-fixed<sub>30yr</sub> baseline (black, solid) and for the detrended baseline (blue). Additionally, the WMO-dashed lines represent the explained variance based on the same fixed<sub>30yr</sub> baseline but detection, with the trend removed after calculating in the annual mean-MHW characteristics (black, dashed) and for the linear baseline (blue) removed.

term trend, cause the strong increase in MHW coverage seen with the WMO baseline. In several years the MHW coverage exceeds 50% without any MHWs occurring at the surface during the same or previous years, independent of the baseline used.

# fixed<sub>30yr</sub> baseline.

Beyond 300 m depth, coherent MHWs occur over layers a few 100 to 1000 m thickness (figure 10a,b). Between 300 and 1200 m only few 1300 m only two short MHWs occur before 2005, but the MHWs coverage continuously exceeds 70% ocean is in a near-permanent MHW state afterwards. This layer is split by a thin layer of nearly zero MHW coverage even after 2005. layer extending from 600 to 1000 m with only intermittent MHWs even after 2005 (figure 10a). When applying the linear baseline detrended baseline (figure 10b), this depth range is divided into three layers (300-600m, the depth ranges between 300 and 800 and 800 to 1200 m seem to be more disconnected 600-1000m and 1000-1300m) with distinct heat content variability and therefore different timing of MHWs. This suggests that the apparent coherence between these depth ranges when applying the WMO-fixed 30 per baseline is mostly caused by a similar temperature trend, while variability on shorter timescales (reflected by the detrended baseline results) has different timings.

From 1200 From 1300 m to 2200 m deptha high MHW coverage is seen until, the water column is occupied by a long lasting MHW before 1985 and nearly no MHWs afterwards (WMO baselinefixed<sub>30yr</sub> baseline; figure 10a). Using the linear detrended baseline, MHWs additionally occur after 2010. This 2010 (figure 10b). Therefore, this depth range is characterised by a long term cooling trend, as well as multi-decadal variability that reached a high phase in the 1980s and 2010s. Below 2200 m (fixed  $30_{yr}$ ) or 2000 m (detrended) intermittent high MHW coverage occurs throughout the entire timeseries. Long-term trends are of minor importance in this depth range, apparent by the similarity between the WMO and linear baselines fixed<sub>30yr</sub> and detrended baselines (figure 10a,b).

In general, the main results derived from the annual mean MHW characteristics are directly represented in the daily MHW coverage. MHWs can extend beyond the mixed layer, but they are not in-phase with surface MHWs and persist longer. Beyond 300 m depth, coherent MHWs occur over layers a few 100 to 1000 m thickness (figure ??c,d). While coherent within the layer, the occurrence of MHWs does not seem to be related across layers, Accordingly, MHWs occur coherently within each of the described layers, but appear unconnected across layers due to different long-term trends and different temperature trends and timing of interannual to decadal variability. Depth levels that show coherent variability are mostly independent of the baseline used. The only notable difference is that the explained variance profile of the MHW coverage in 450 m depth shows a secondary maximum at 1100 m depth when the WMO baseline is applied. This secondary peak is missing for the linear baseline. As explained above this is caused by similar trends, but different timing of interannual to decadal variability. It is important to note that the described temporal changes in MHW characteristics apply to the entire these MHWs cover most of the archipelago region and are not related to processes that only occur close to the islands for example.

#### As none of the depth levels

## 3.4.3 Drivers of individual MHWs

Our results described in the previous paragraphs show that the temporal evolution of MHW metrics and the occurrence of individual MHWs below 300 m show any hint on connections are not linked to the surface forcing. Instead, oceanic processes

what sets the vertical extend of coherent MHWs<del>we</del> calculate a heat budget for the Cape Verde archipelago. As expected, the MHW coverage closely follows the heat content integrated over the region (figure ??a). The heat content shows pronounced interannual variability at the surface, a long-term positive trend between 100 and 1400 m depth and a negative trend below 1400 m that becomes small below 3000 m depth. This structure is also reflected in the mean heat content change between 1980 and 2022, which itself is caused by a residual of much larger heat fluxes (figure ??c). At the surface the ocean gains heat trough the air-sea heat fluxes. This heat is transferred to deeper levels by mixing, which is part of , we analyse the contribution of different heat budget terms to the heat content anomalies associated with individual MHWs.

Within the mixed layer, MHWs detected with the detrended baseline are almost exclusively driven by the air-sea heat exchange and subsequent downward mixing (figure 10d). Mixing is included in the residual flux. At the base that has a maximum in the mixed layer. Approximately at the annual mean depth of the mixed layer the vertical heat advection term shows a positive maximum, indicating that the ocean gains heat through vertical heat convergence in this depth range. The horizontal convergence of the heat transport counteracts the vertical transport at all depth levels and cools the ocean between 50 in 70 m, MHWs are primarily driven by the vertical heat transport ( $OHT_{vv}$ ). Between 100 and 200 m. Between 200 and 600 m the opposite is true. Here upwelling of colder water from depth leads to a mean cooling, balanced by a horizontal convergence of heat. Below horizontal ocean heat transport (OHT) and  $OHT_{vv}$  both contribute to MHW events. The OHT contribution decreases below 600 mthe mean fluxes become very small compared to their near-surface values. While the terms are balanced in the mixed layer, there is small residual that leads to a heat content change (and causes a linear trend in the OHC, when integrated over time). Even though the sign of the vertical and horizontal heat convergence alternate with depth, the residual grows until 1200 m and becomes negative at 1500 m. This suggests that it is not a single mechanism that leads to the trend, such that below 1000 m MHWs are exclusively driven by the vertical heat transport. The contribution of the different heat budget terms is similar when MHWs are detected with the fixed  $_{30yr}$  baseline, but the horizontal heat transport has a weaker contribution to MHWs between 300 and 600 m (figure 10c).

When removing the impact of the linear trend, MHWs typically occur following periods of sustained positive heat content change (figure ??b). However, this must not always be true since a strong heat content increase will not result in a MHW, if the ocean is anomalously cold before. As for the MHW coverage, the heat content change does not vary in phase across the entire water column. The layers in which MHWs occur at the same time are strongly related to the processes that cause the heat content to vary at different depths (figure ??d, ??c,d). Within the mixed layer (above 50 m) heat content variability is dominated by vertical mixing of heat that is gained trough the surface. At the base of the mixed layer most of the heat content variance is explained by vertical The different contribution of the heat advection, budget terms in different depth, together with the depth structure of these terms itself, can explain the vertical characteristics of MHWs (detrended baseline) in the region. Within the first 100 m MHWs are governed by the surface heat flux and vertical displacements of the thermocline (figure 10). Between 100 and 300 m, MHWs are caused by the interaction of positive vertical and horizontal heat transport anomalies. Although MHWs in the depth range between 300 and 700 m lateral and vertical heat advection explain similar fractions of the

variance, while between 800 and 1300 m the vertical term reaches 70% explained variance. Between 1600 and 2400 m it is the horizontal heat convergence that dominates heat content variability. Below 2400 m it is again the vertical advection that becomes more important.

The different relative importance of the vertical and lateral heat convergence are in turn 600 m are also driven by both, OHT and  $OHT_w$ , the horizontal transport shows distinct variability in this depth range (figure 11a). The horizontal temperature difference between the interior and exterior of the Cape Verde archipelago region strongly decreases in this depth range (figure 11c). This is related to the vertical structure of the currents and temperature field. In the mixed layer horizontal currents are strong, but the temperature transition from North Atlantic Central Water, with a pronounced horizontal gradient across the domain boundary is small. At the base of the mixed layer, the speed of horizontal currents is already only a small fraction of their surface value (although still much larger than vertical velocities in absolute terms). Here the vertical temperature gradient reaches a maximum, explaining the high importance of vertical advection. Between 300 and 700 m Cape Verde Frontal Zone, to Intermediate Water with a much smaller horizontal temperature gradient (Liu and Tanhua, 2021; Zenk et al., 1991).

Between 600 m and 1000 m depth the horizontal temperature gradient reaches a maximum, but also the vertical speed is comparably large. Therefore both, horizontal and vertical heat transports, contribute to heat content variations. Below 700 m the horizontal temperature gradient is small again, while vertical velocities and vertical gradients still reach more than 10% of their maximum value and therefore have a larger impact. Note that around 800 m depth, which appears as a level with distinct variability as explained above, the mean horizontal temperature gradient becomes negative. Thus a vertically coherent circulation change will lead to opposite signed heat content changes in different depths. Between 1600 m and 2300 m the vertical temperature gradient reaches near-zerodifference reverses sign, i.e. the Cape Verde archipelago region is warmer than the surrounding (figure 11c). The temperature difference is comparably small, but the horizontal heat transport still contributes to events (figure 10d). Due to the reversed sign, a barotropic velocity change is expected to cause opposing heat transport anomalies between 300 and 600 and 600 and 1000 m depth. This is where the contribution of vertical advection becomes negligible. Below 2700 m depththe vertical temperature gradient grows slightly. In the presence of comparably large vertical velocities this causes the vertical heat transport to be again important for the generation of heat content changes. Therefore, the changing relative importance of the vertical heat transport (figure ??d) is mostly caused by changes indeed often visible in figure 11a. As a consequence, MHWs typically do not occur at the same time.

Below 1000 m depth, the vertical transport of heat dominates the development of MHWs (figure 10d). Nevertheless, a break in the vertical temperature gradient. In 1000 and 3000 coherence occurs around 1300 m depth, where the vertical heatflux has a major contribution to heat content changes, enhanced vertical gradients can be seen across the entire domain (not shown). Thus, the large-scale temperature stratification rather than local effects close to the islands set the vertical structure of coherent MHWs.

In conclusion,MHWs in temperature gradient shows a minimum (figure 11d). Consistently, vertical heat transport anomalies show a minimum as well (figure 11b). Below 2000 m, the vertical gradient and vertical heat transport increase again and lead to intermittent short MHW events (figure 11b,d). Different long-term temperature trends in different depths slightly modify the exact transition between the described layers of coherent MHW occurrence when the fixed<sub>30µr</sub> baseline is used. Nevertheless,

the Cape Verde archipelago are related to surface fluxes in the top 100 m of the ocean. This includes the mixed layer, but MHWs can extend beyond the mixed layer through subduction. Even though MHWs are detected at individual grid points and depth levels, coherent structures can be identified. The depth range over which coherent MHWs occur are closely related to the vertical and horizontal described vertical structure of the temperature field. Both, horizontal and vertical heat advection play amajor role. Changes in their relative importance, which are primarily linked to vertical changes in the large-scale temperature stratification, set the depth ranges over which coherent MHWs occur. If the linear baseline is applied, the detected MHWs are related to pentadal to decadal variability at depth, while applying a fixed (WMO) baseline results in a combination of both, pentadal to decadal variability and the long-term temperature trend. Nevertheless, the different baselines yield similar depth ranges, over which MHWs occur at the same time. The described processes act over the entire archipelago and are not related to processes that occur along the island slopes for example, heat budget terms remains visible (figure 10a,c). Note that linear trend in temperature (and heat content) itself results from a small imbalance in the time mean fluxes shown in figure 11a,b.

## 4 Discussion and Conclusion

## 4.1 Detecting MHWs at depth

In this study we have analysed the characteristics of temperature extremes (MHWs) throughout the entire Atlantic Ocean. By using a hierarchy of ocean model grids we identified the impact of the horizontal resolution, ocean dynamics and different baselines on the derived MHW statistics.

We find that interpolation of the temperature from a 1/20° to a 1/4° grid does not impact the derived statistics. This is important for the interpretation of model results, but has further implications for the detection of MHWs from gridded observational products. The reason for this result is likely that MHWs do not occur at isolated grid points, but almost always cover an area that is larger than the grid resolution of the datasets (mostly 1/20° to 1/4°). Thus, as long as the target grid size is still small enough to capture the typical extend of MHWs in the region, it is expected that the horizontal resolution of the temperature dataset does not affect the MHW statistics. This would also mean that the interpolation of high-resolution along track data onto a regular, coarser grid does not impact the detection of MHWs.

However, statistics change if the horizontal resolution of the grid affects the dynamic scales resolved in a model. The coarser resolution model with otherwise same surface forcing and initial conditions generally overestimates the duration and underestimates the frequency and intensity of MHWs. This result is consistent with a coupled model study conducted by Pilo et al. (2019). Coarse horizontal resolution is often mentioned as a limiting factor in MHW studies (e.g. Hövel et al., 2022). In the presence of highly variable currents and at mid-depths (100 - 3000 m) differences between the high and coarse resolution configurations can be substantial. This means that studying the impact of MHWs on deep ecosystems requires models with sufficiently high resolution, in particular along the continental slopes. Differences are overall small at the surface, except for the NAC regionGulf Stream and North Atlantic Current (NAC) regions, but the discrepancy is expected to be larger for coarser 1/2° and 1° models (Pilo et al., 2019).

While Hobday et al. (2016) mention that horizontal and temporal resolution of the temperature dataset are important, we argue that the horizontal resolution (in certain limits that are probably related to the typical extend of MHWs) plays a minor role, as long as the dynamics represented in the dataset are the same. This is highly advantageous since MHWs can be detected equivalently on the native model grid without interpolation, or at a coarser resolution, whatever option is less computational expensive. This simplifies the comparison of different satellite products and models. As a result, the applied two-way nesting applied here provides provided a convenient tool to produce a manageable dataset containing mesoscale effects. Note that detecting MHWs on the The computational resources needed for this study would have been much higher, if MHWs had to be detected on the high-resolution grid. Detecting MHWs on the high resolution grid requires approximately 25 times more computing time and significantly more resources for analysis and storage. Daily MHW statistics for 43 years on all 46 depth levels on the coarse grid (only the domain covered by the nest) take up 90 GB of storage, while it is 700 GB on the high-resolution grid. The amount of data that needs to be processed is even higher, since the output file size is strongly reduced by compression.

The choice of a suitable baseline is widely discussed in the current literature and depends on the scientific question (Amaya et al., 2023b) (Amaya et al., 2023b; Smith et al., 2025), Modelled temperature trends can strongly differ across experiments, for example dependent on the time of the model spin-up (adjustment to from initial conditions). While trends are robust in approximately the top 100 m, they strongly depend on the time of the model change with the model's spin-up time at greater depths. Trends at depth vary in magnitude and even have opposite signs in our model experiments that just differ in the initial conditions. As a consequence, the WMO and fixedbaselines are not applicable fixed baselines (e.g. the fixed<sub>30w</sub> and fixed<sub>43w</sub> baselines used 680 here) do not yield reasonable results below approximately 100 m depth in the presence of strong model drift. This is Instead, MHW statistics are dominated by trends that arise from the models adjustment from the initial conditions, rather than trends related to the surface forcing or intrinsic oceanic variability. The choice of the baseline is then not only a question of interpretation, but the slow adjustment of the deep circulation does not allow for any meaningful interpretation of the results. We have only investigated this in only one model configuration here, but model drift is common to nearly all forced and coupled 685 models (e.g. Tsujino et al., 2020). As a consequence, if the aim is to include multi-decadal trends in the MHWs statistics at beyond approximately 100 m depth, a model spin-up with sufficient time to allow the deep-ocean sub-surface to equilibrate is needed(around 200 years in VIKING20X). For the abyssal basins with water. It is not possible to provide a universally applicable recommendation on the exact spin-up time or procedure. We have only tested the spin-up strategy recommended by 690 the OMIP-II protocol (Tsujino et al., 2020), but other strategies exist as well. Still our results show that the required spin-up time will depend on the depth. For the surface it was shown that a 22 year long spin-up (1958-1980 in the 1st cycle) is sufficient. Mid-depth water masses do not show a strong temperature drift after the 2nd cycle (124 years; not shown), while abyssal water masses at depths beyond 5000 m the comparison of the WMO and linear baselines suggest that an even longer have not fully adjusted even after 300 years of spin-upis necessary. Such a long spin-up is often not feasible at this resolution. Nevertheless, it was shown that at the surface, where both cycles are dominated by the surface forcing, similar results are obtained. Studies that . Therefore, studies that focus on near-surface MHWs get away with using a much shorter short spin-up of only a few years, as the ocean model typically stabilises quickly. Studies that focus on the deeper ocean will need a much longer spin-up time with probably more than 300 years for abyssal basins. Such long spin-ups are often not feasible at the resolution used in this study. Additional constraints to reduce model drift, such as the assimilation of observations, could also alleviate the problem, while at the same time acknowledging the fact that deep observations are sparse. Ocean reanalysis was successfully used by Fragkopoulou et al. (2023) to study MHWs at depth. Nevertheless, frequent and widespread observations typically exist only in the top 1000 m and thus also reanalysis products should be treated with caution in the deep ocean. Furthermore, a downside of many assimilation techniques, in particular of nudging, is that they may violate conservation laws (Zeng and Janjić, 2016; Janjić et al., 2014) and thus studying the drivers of MHWs is problematic in such datasets due to spurious sources and sinks of heat. With a detrended baseline, statistics are not identical, but similar in the presence (1st cycle) and absence (6th cycle) of model drift. Studies applying a detrended baseline to focus in interannual to decadal variability are therefore possible with a relatively short model spin-up (roughly 20 years) as well.

#### 4.2 Characteristics, drivers and trends








From their definition it immediately follows that MHWs need to occur everywhere (disregarding the condition that the temperature threshold must be exceeded for 5 consecutive days). Therefore, the pure observation that MHWs occur throughout the entire water column is not surprising. Still, the detection of MHWs is a useful tool to comprehensively study the characteristics of temperature variability and how it changes in time. In the upper ocean temperatures cover a large range the temperature is highly variable (large variance) and anomalously high temperatures occur for relatively short times. In the deep ocean temperature anomalies associated with MHWs are smaller and occur on longer timescales compared to the surface. Along the continental slope, MHWs with maximum intensities in the order of 1°C do occur. Given that the temperature tolerance of deep-sea species like cold-water corals is expected to be around 4°C (Morato et al., 2020), this could have major impacts for ecosystems that are already close to their upper temperature limit. In the abyssal ocean MHWs MHW intensities do not exceed 0.1°C. Whether such low temperature variations have any impact is yet to be investigated. In general, even small temperature anomalies could have an impact, if they are sustained for sufficiently long times. At the same time, ecosystems may have adapted to large temperature variability in regions where MHWs occur very frequently and thus do not represent rare events. The aim of this study is to provide a comprehensive understanding about the characteristics of temperature variability throughout the entire Atlantic which, when combined with biological information, will help to identify deep ecosystems that may be vulnerable to MHWs and changes in their characteristics.

Overall, our results highlight the importance of the ocean circulation on the development and characteristics of MHWs. By comparing two model configurations that only differ by their horizontal resolution, we find that mesoscale dynamics change the frequency, duration and maximum intensity of MHWs, in particular at depth. In agreement with Großelindemann et al. (2022); Zhang et al. (2023); Elzahaby and Schaeffer (2019); Wyatt et al. (2023) and Wu and He (2024) this is partly caused by mesoscale features such as eddies and meanders themselves. Additionally, indirect effects, such as changes in current structure, strength, mixed layer dynamics, and vertical as well as horizontal temperature gradients contribute to differences between the eddy-permitting and eddy-rich configurations in our study. Highly variable currents, such as the NAC and DWBC are related to the occurrence of short, but frequent MHWs in agreement with Fragkopoulou et al. (2023). Also the Mediterranean Sea Outflow, deep Deep convection in the Labrador Sea and upwelling were found to strongly influence the occurrence and characteristics of MHWs at depth. As the surface forcing can be ruled out as the cause, this Additionally, strong vertical temperature gradients at mid-depth (300-1500 m depth), for example between warm Mediterranean Outflow Water and colder North Atlantic Deep Water, lead to regionally more intense MHWs. These MHWs are not be necessarily caused by variability in the properties of the water masses themselves, but rather by a vertical displacement of isotherms through internal waves or wind driven up-/downwelling. Overall, the pronounced differences between the configurations, despite using the same atmospheric forcing, points to a major influence of ocean dynamics on the occurrence of MHWsat depth, but also at the surface. In most regions of the Atlantic characteristics of MHWs.

In both model experiments (VIKING20X-1st and VIKING20X-6th) analysed here, the impact of the local surface forcing on MHWs is limited to approximately the top 100 m throughout the Atlantic, with some minor regional differences. Consistent with other studies (Xu et al., 2022; Oliver et al., 2018; Chiswell, 2022) we find a positive trend in all MHW characteristics when applying a fixed baseline at the surface. Although not related to the surface forcing, but changes in ocean dynamics, positive trends can be seen in most regions until a depth of roughly 1000 m. Between 1000 and 4000 m negative trends prevail in the model. Although model trends are always connected to a large uncertainty, in particular in the deep ocean, this result suggests that many deep ecosystems experienced a decrease in extreme (positive) temperatures frequency, duration and intensity of MHWs over time. At the very least the model shows that even though trends are clearly positive at the surface, it can not be expected that marine heatwaves increased over time throughout the water column.

Based on a more detailed investigation. A detailed study of MHWs in the Cape Verde archipelago we find that for shows that the depth of the mixed layer marks a transition zone for both the long-term trend, but also for and variability on shorter timescales, the depth of approximately 100 m marks a transition zone. This is caused by very different characteristics within and below. MHWs in the mixed layer, which is consistent are almost exclusively driven by the surface heat flux. Consistent with results obtained by Scannell et al. (2020) and Amaya et al. (2023a), MHWs within and below the mixed layer have very different characteristics. In agreement with these studies, surface forced MHWs can be detrained from the seasonally varying mixed layer. If they are subducted below the annual maximum mixed layer, they can persist for several years, otherwise they may re-emerge at the surface while MHWs closer to the surface are typically much shorter.






Below approximately 100 - 300 m depth, the surface forcing does not affect MHWs in most regions. Here vertical and lateral ocean heat transports control the heat budget and therefore the occurrence of the Cape Verde archipelago. Instead, vertical and horizontal ocean heat transport anomalies drive MHWs. While MHWs are detected independently at each model level, coherent vertical changes in heat content lead to vertically coherent MHWs. Depth levels that show coherent MHW events are mostly independent of the baseline used and span depth ranges of a few 100 to 1000 m. The vertical extent of these depth ranges can be directly linked to the vertical structure of the MHWs' physical drivers.

Below the mixed layer and above 1000 m depth, horizontal and vertical heat transport both contribute to MHWs. In deeper layers, slow horizontal currents and a vanishing horizontal temperature gradient lead to a dominance of the vertical transport. Different temperature trends in different depths can slightly modify the depth ranges in which MHWs. For the Cape Verde archipelago the vertical structure of the temperature field sets the depth range over which vertically coherent MHWs occur. It is an important result itself that vertically coherent MHWs are detected at all, even though the definition is applied to every grid point individually. Although this was studied explicitly only occur coherently. Nevertheless, the impact of changing ocean heat transports on shorter timescales is apparent for both, the detrended and fixed 30 yr baselines. It is important to note the described processes (e.g. changes in the vertical heat transport) act over the entire archipelago and are not related to processes that occur along the island slopes for example.

Although this detailed analysis was only carried out for the Cape Verde archipelago, the Atlantic wide statistics suggest that similar mechanisms occur throughout most of the basin. As a result, our study strongly supports the conclusions of Sun

et al. (2023); Zhang et al. (2023); Elzahaby and Schaeffer (2019); Schaeffer and Roughan (2017) and Wyatt et al. (2023) that measuring temperature at the surface alone yields no information on extreme temperature events below the mixed layer. Conversely, studying MHWs at depth will require detailed knowledge of ocean dynamics. This includes vertical velocities that are very small compared to horizontal velocities, but can be very important due to larger vertical than horizontal temperature gradients.



In conclusion, this study presents results of a single model simulation, but the main results are consistent with various other publications as described above (e.g. Fragkopoulou et al., 2023; Wu and He, 2024; Großelindemann et al., 2022). The mean characteristics at the surface and at depth are qualitatively and quantitatively similar to the ocean reanalysis based study of Fragkopoulou et al. (2023). As direct observations for a larger domain at daily resolution are not available below the surface, studies of MHWs at depth will have to rely on models in the foreseeable future. This study provides valuable information about the characteristics of MHWs at depth and how they are related to ocean dynamics, as well as a potential pitfalls challenges when detecting deep MHWs in models. Additionally, it provides a unique dataset to launch investigations on the impact of MHWs on subsurface ecosystems.

# 790 Appendix A: Appendix A

Code and data availability. The full MHW detection output for the 6th cycle of VIKING20X is available trough GEOMAR at https://hdl.handle.net/20.500.12085/49913d6b-4c70-43cb-9d3c-b4b73b0b8291 (Schulzki et al., 2025a). Additional data and material that support the findings of this study are available through GEOMAR at https://hdl.handle.net/20.500.12085/a3279a60-e9ef-437f-bd34-c3e156181e98 (Schulzki et al., 2025b)

Author contributions. AB initiated and designed the study. FUS performed the numerical model experiments. All authors contributed to the design of the analysis, which TS performed including the marine heatwave detection. TS wrote the manuscript draft which all authors jointly iterated.

Competing interests. The authors declare that they have no conflict of interest.

Acknowledgements. The study was supported by the European Union's Horizon 2020 research and innovation program under Grant Agree-ment 818123 (iAtlantic) and by the Federal Ministry of Education and Research (BMBF) through the project METAscales (Grant No. 03F0955J), as well as the northern German states within the scope of the German Marine Research Alliance (DAM) mission mareXtreme. The authors gratefully acknowledge the Earth System Modelling Project (ESM) for funding this work by providing computing time on the ESM partition of the supercomputer JUWELS at the Juelich Supercomputing center (JSC). We thank the three reviewers of this study for their detailed comments and suggestions.

- Amaya, D. J., Jacox, M. G., Alexander, M. A., Scott, J. D., Deser, C., Capotondi, A., and Phillips, A. S.: Bottom marine heatwaves along the continental shelves of North America, Nature Communications, 14, 1038, https://doi.org/10.1038/s41467-023-36567-0, 2023a.
- Amaya, D. J., Jacox, M. G., Fewings, M. R., Saba, V. S., Stuecker, M. F., Rykaczewski, R. R., Ross, A. C., Stock, C. A., Capotondi, A., Petrik, C. M., Bograd, S. J., Alexander, M. A., Cheng, W., Hermann, A. J., Kearney, K. A., and Powell, B. S.: Marine heatwaves need clear definitions so coastal communities can adapt, Nature, 616, 29–32, https://doi.org/10.1038/d41586-023-00924-2, 2023b.
- Arístegui, J., Barton, E. D., Álvarez Salgado, X. A., Santos, A. M. P., Figueiras, F. G., Kifani, S., Hernández-León, S., Mason, E., Machú, E., and Demarcq, H.: Sub-regional ecosystem variability in the Canary Current upwelling, Progress in Oceanography, 83, 33–48, https://doi.org/10.1016/j.pocean.2009.07.031, 2009.
- Barnier, B., Madec, G., Penduff, T., Molines, J.-M., Treguier, A.-M., Le Sommer, J., Beckmann, A., Biastoch, A., Böning, C., Dengg, J.,
  Derval, C., Durand, E., Gulev, S., Remy, E., Talandier, C., Theetten, S., Maltrud, M., McClean, J., and Cuevas, B.: Impact of partial steps and momentum advection schemes in a global ocean circulation model at eddy-permitting resolution, Ocean Dynamics, 56, 543–567, https://doi.org/10.1007/s10236-006-0082-1, 2006.
  - Behrens, E., Fernandez, D., and Sutton, P.: Meridional Oceanic Heat Transport Influences Marine Heatwaves in the Tasman Sea on Interannual to Decadal Timescales, Frontiers in Marine Science, 6, https://doi.org/10.3389/fmars.2019.00228, 2019.
- Berthou, S., Renshaw, R., Smyth, T., Tinker, J., Grist, J. P., Wihsgott, J. U., Jones, S., Inall, M., Nolan, G., Berx, B., Arnold, A., Blunn, L. P., Castillo, J. M., Cotterill, D., Daly, E., Dow, G., Gómez, B., Fraser-Leonhardt, V., Hirschi, J. J.-M., Lewis, H. W., Mahmood, S., and Worsfold, M.: Exceptional atmospheric conditions in June 2023 generated a northwest European marine heatwave which contributed to breaking land temperature records, Communications Earth & Environment, 5, 287, https://doi.org/10.1038/s43247-024-01413-8, 2024.
- Biastoch, A., Schwarzkopf, F. U., Getzlaff, K., Rühs, S., Martin, T., Scheinert, M., Schulzki, T., Handmann, P., Hummels, R., and Böning, C. W.: Regional imprints of changes in the Atlantic Meridional Overturning Circulation in the eddy-rich ocean model VIKING20X, Ocean Sci., 17, 1177–1211, https://doi.org/10.5194/os-17-1177-2021, publisher: Copernicus Publications, 2021.
  - Böning, C. W., Wagner, P., Handmann, P., Schwarzkopf, F. U., Getzlaff, K., and Biastoch, A.: Decadal changes in Atlantic overturning due to the excessive 1990s Labrador Sea convection, Nature Communications, 14, 4635, https://doi.org/10.1038/s41467-023-40323-9, 2023.
  - Chen, K., Gawarkiewicz, G., and Yang, J.: Mesoscale and Submesoscale Shelf-Ocean Exchanges Initialize an Advective Marine Heatwave, Journal of Geophysical Research: Oceans, 127, e2021JC017 927, https://doi.org/10.1029/2021JC017927, publisher: John Wiley & Sons, Ltd. 2022.
  - Chiswell, S. M.: Global Trends in Marine Heatwaves and Cold Spells: The Impacts of Fixed Versus Changing Baselines, Journal of Geophysical Research: Oceans, 127, e2022JC018757, https://doi.org/10.1029/2022JC018757, publisher: John Wiley & Sons, Ltd, 2022.
- Cropper, T. E., Hanna, E., and Bigg, G. R.: Spatial and temporal seasonal trends in coastal upwelling off Northwest Africa, 1981–2012, Deep Sea Research Part I: Oceanographic Research Papers, 86, 94–111, https://doi.org/10.1016/j.dsr.2014.01.007, 2014.
  - Debreu, L., Vouland, C., and Blayo, E.: AGRIF: Adaptive grid refinement in Fortran, Computers & Geosciences, 34, 8–13, https://doi.org/10.1016/j.cageo.2007.01.009, 2008.
  - Elzahaby, Y. and Schaeffer, A.: Observational Insight Into the Subsurface Anomalies of Marine Heatwaves, Frontiers in Marine Science, 6, https://doi.org/10.3389/fmars.2019.00745, 2019.

- Elzahaby, Y., Schaeffer, A., Roughan, M., and Delaux, S.: Oceanic Circulation Drives the Deepest and Longest Marine Heatwaves in the East Australian Current System, Geophysical Research Letters, 48, e2021GL094785, https://doi.org/10.1029/2021GL094785, publisher: John Wiley & Sons, Ltd, 2021.
- Fox, A. D., Handmann, P., Schmidt, C., Fraser, N., Rühs, S., Sanchez-Franks, A., Martin, T., Oltmanns, M., Johnson, C., Rath, W., Holliday, N. P., Biastoch, A., Cunningham, S. A., and Yashayaev, I.: Exceptional freshening and cooling in the eastern subpolar North Atlantic caused by reduced Labrador Sea surface heat loss, Ocean Sci., 18, 1507–1533, https://doi.org/10.5194/os-18-1507-2022, publisher: Copernicus Publications, 2022.
  - Fragkopoulou, E., Sen Gupta, A., Costello, M. J., Wernberg, T., Araújo, M. B., Serrão, E. A., De Clerck, O., and Assis, J.: Marine biodiversity exposed to prolonged and intense subsurface heatwaves, Nature Climate Change, 13, 1114–1121, https://doi.org/10.1038/s41558-023-01790-6, 2023.
- Gawarkiewicz, G., Chen, K., Forsyth, J., Bahr, F., Mercer, A. M., Ellertson, A., Fratantoni, P., Seim, H., Haines, S., and Han, L.: Characteristics of an Advective Marine Heatwave in the Middle Atlantic Bight in Early 2017, Frontiers in Marine Science, 6, https://www.frontiersin.org/journals/marine-science/articles/10.3389/fmars.2019.00712, 2019.

- Gelderloos, R., Katsman, C. A., and Drijfhout, S. S.: Assessing the Roles of Three Eddy Types in Restratifying the Labrador Sea after Deep Convection, Journal of Physical Oceanography, 41, 2102–2119, https://doi.org/10.1175/JPO-D-11-054.1, place: Boston MA, USA Publisher: American Meteorological Society, 2011.
- Goes, M., Dong, S., Foltz, G. R., Goni, G., Volkov, D. L., and Wainer, I.: Modulation of Western South Atlantic Marine Heatwaves by Meridional Ocean Heat Transport, Journal of Geophysical Research: Oceans, 129, e2023JC019715, https://doi.org/10.1029/2023JC019715, publisher: John Wiley & Sons, Ltd, 2024.
- Großelindemann, H., Ryan, S., Ummenhofer, C. C., Martin, T., and Biastoch, A.: Marine Heatwaves and Their Depth Structures on the Northeast U.S. Continental Shelf, Frontiers in Climate, 4, 2022.
  - Guo, X., Gao, Y., Zhang, S., Wu, L., Chang, P., Cai, W., Zscheischler, J., Leung, L. R., Small, J., Danabasoglu, G., Thompson, L., and Gao, H.: Threat by marine heatwaves to adaptive large marine ecosystems in an eddy-resolving model, Nature Climate Change, 12, 179–186, https://doi.org/10.1038/s41558-021-01266-5, 2022.
- Hobday, A. J., Alexander, L. V., Perkins, S. E., Smale, D. A., Straub, S. C., Oliver, E. C., Benthuysen, J. A., Burrows, M. T., Donat, M. G.,
   Feng, M., Holbrook, N. J., Moore, P. J., Scannell, H. A., Sen Gupta, A., and Wernberg, T.: A hierarchical approach to defining marine heatwaves, Progress in Oceanography, 141, 227–238, https://doi.org/10.1016/j.pocean.2015.12.014, 2016.
  - Holbrook, N. J., Scannell, H. A., Sen Gupta, A., Benthuysen, J. A., Feng, M., Oliver, E. C. J., Alexander, L. V., Burrows, M. T., Donat, M. G., Hobday, A. J., Moore, P. J., Perkins-Kirkpatrick, S. E., Smale, D. A., Straub, S. C., and Wernberg, T.: A global assessment of marine heatwaves and their drivers, Nature Communications, 10, 2624, https://doi.org/10.1038/s41467-019-10206-z, 2019.
- Hoving, H. J. T., Neitzel, P., Hauss, H., Christiansen, S., Kiko, R., Robison, B. H., Silva, P., and Körtzinger, A.: In situ observations show vertical community structure of pelagic fauna in the eastern tropical North Atlantic off Cape Verde, Scientific Reports, 10, 21798, https://doi.org/10.1038/s41598-020-78255-9, 2020.
  - Huang, B., Liu, C., Banzon, V., Freeman, E., Graham, G., Hankins, B., Smith, T., and Zhang, H.-M.: Improvements of the Daily Optimum Interpolation Sea Surface Temperature (DOISST) Version 2.1, Journal of Climate, 34, 2923–2939, https://doi.org/10.1175/JCLI-D-20-0166.1, place: Boston MA, USA Publisher: American Meteorological Society, 2021.
  - Hövel, L., Brune, S., and Baehr, J.: Decadal Prediction of Marine Heatwaves in MPI-ESM, Geophysical Research Letters, 49, e2022GL099 347, https://doi.org/10.1029/2022GL099347, publisher: John Wiley & Sons, Ltd, 2022.

- Janjić, T., McLaughlin, D., Cohn, S. E., and Verlaan, M.: Conservation of Mass and Preservation of Positivity with Ensemble-Type Kalman Filter Algorithms, Monthly Weather Review, 142, 755–773, https://doi.org/10.1175/MWR-D-13-00056.1, place: Boston MA, USA Publisher: American Meteorological Society, 2014.
  - Kaboth-Bahr, S., Bahr, A., Stepanek, C., Catunda, M. C. A., Karas, C., Ziegler, M., Garcia-Gallardo, A., and Grunert, P.: Mediterranean heat injection to the North Atlantic delayed the intensification of Northern Hemisphere glaciations, Communications Earth & Environment, 2, 158, https://doi.org/10.1038/s43247-021-00232-5, 2021.
- Krüger, J., Kjellsson, J., Kedzierski, R. P., and Claus, M.: Connecting North Atlantic SST Variability to European Heat Events over the Past Decades, Tellus A: Dynamic Meteorology and Oceanography, https://doi.org/10.16993/tellusa.3235, 2023.
  - Lee, T., Fukumori, I., and Tang, B.: Temperature Advection: Internal versus External Processes, Journal of Physical Oceanography, 34, 1936–1944, https://doi.org/10.1175/1520-0485(2004)034<1936:TAIVEP>2.0.CO;2, place: Boston MA, USA Publisher: American Meteorological Society, 2004.
- Liu, M. and Tanhua, T.: Water masses in the Atlantic Ocean: characteristics and distributions, Ocean Science, 17, 463–486, https://doi.org/10.5194/os-17-463-2021, 2021.
  - Locarnini, R. A., Mishonov, A. V., Antonov, J. I., Boyer, T. P., Garcia, H. E., Baranova, O. K., Zweng, M. M., Paver, C. R., Reagan, J. R., Johnson, D. R., Hamilton, M., Seidov, 1948, D., and Levitus, S.: World ocean atlas 2013. Volume 1, Temperature, https://doi.org/10.7289/V55X26VD, 2013.
- Madec, G.: NEMO ocean engine, Note du P\^ole de mod{\'e}lisation, Institut Pierre-Simon Laplace (IPSL), France, No 27, ISSN No 1288-895 1619, 2016.
  - Maldonado, M., Aguilar, R., Bannister, R. J., Bell, J. J., Conway, K. W., Dayton, P. K., Díaz, C., Gutt, J., Kelly, M., Kenchington, E. L. R., Leys, S. P., Pomponi, S. A., Rapp, H. T., Rützler, K., Tendal, O. S., Vacelet, J., and Young, C. M.: Sponge Grounds as Key Marine Habitats: A Synthetic Review of Types, Structure, Functional Roles, and Conservation Concerns, in: Marine Animal Forests: The Ecology of Benthic Biodiversity Hotspots, edited by Rossi, S., Bramanti, L., Gori, A., and Orejas, C., pp. 145–183, Springer International Publishing, Cham, ISBN 978-3-319-21012-4, https://doi.org/10.1007/978-3-319-21012-4 24, 2017.
  - Marzinelli, E. M., Williams, S. B., Babcock, R. C., Barrett, N. S., Johnson, C. R., Jordan, A., Kendrick, G. A., Pizarro, O. R., Smale, D. A., and Steinberg, P. D.: Large-Scale Geographic Variation in Distribution and Abundance of Australian Deep-Water Kelp Forests, PLOS ONE, 10, e0118 390, https://doi.org/10.1371/journal.pone.0118390, publisher: Public Library of Science, 2015.

- Morato, T., Gonzalez-Irusta, J.-M., Dominguez-Carrio, C., Wei, C.-L., Davies, A., Sweetman, A. K., Taranto, G. H., Beazley, L., GarciaAlegre, A., Grehan, A., Laffargue, P., Murillo, F. J., Sacau, M., Vaz, S., Kenchington, E., Arnaud-Haond, S., Callery, O., Chimienti, G., Cordes, E., Egilsdottir, H., Freiwald, A., Gasbarro, R., Gutierrez-Zarate, C., Gianni, M., Gilkinson, K., Wareham Hayes, V. E., Hebbeln, D., Hedges, K., Henry, L.-A., Johnson, D., Koen-Alonso, M., Lirette, C., Mastrototaro, F., Menot, L., Molodtsova, T., Duran Munoz, P., Orejas, C., Pennino, M. G., Puerta, P., Ragnarsson, S. A., Ramiro-Sanchez, B., Rice, J., Rivera, J., Roberts, J. M., Ross, S. W., Rueda, J. L., Sampaio, I., Snelgrove, P., Stirling, D., Treble, M. A., Urra, J., Vad, J., van Oevelen, D., Watling, L., Walkusz, W., Wienberg, C.,
  Woillez, M., Levin, L. A., and Carreiro-Silva, M.: Climate-induced changes in the suitable habitat of cold-water corals and commercially
- Woillez, M., Levin, L. A., and Carreiro-Silva, M.: Climate-induced changes in the suitable habitat of cold-water corals and commercially important deep-sea fishes in the North Atlantic, Global Change Biology, 26, 2181–2202, https://doi.org/10.1111/gcb.14996, publisher: John Wiley & Sons, Ltd, 2020.
- Oliver, E. C., Benthuysen, J. A., Darmaraki, S., Donat, M. G., Hobday, A. J., Holbrook, N. J., Schlegel, R. W., and Sen Gupta, A.: Marine Heatwaves, Annual Review of Marine Science, 13, 313–342, https://doi.org/https://doi.org/10.1146/annurev-marine-032720-095144, publisher: Annual Reviews Type: Journal Article, 2021.

- Oliver, E. C. J., Donat, M. G., Burrows, M. T., Moore, P. J., Smale, D. A., Alexander, L. V., Benthuysen, J. A., Feng, M., Sen Gupta, A., Hobday, A. J., Holbrook, N. J., Perkins-Kirkpatrick, S. E., Scannell, H. A., Straub, S. C., and Wernberg, T.: Longer and more frequent marine heatwaves over the past century, Nature Communications, 9, 1324, https://doi.org/10.1038/s41467-018-03732-9, 2018.
- Petrelli, P.: XMHW: Xarray based code to identify Marine HeatWave events and their characteristics 920 https://doi.org/10.5281/zenodo.7668235, 2023.
  - Pilo, G. S., Holbrook, N. J., Kiss, A. E., and Hogg, A. M.: Sensitivity of Marine Heatwave Metrics to Ocean Model Resolution, Geophysical Research Letters, 46, 14604–14612, https://doi.org/10.1029/2019GL084928, publisher: John Wiley & Sons, Ltd, 2019.
  - Qiu, Z., Qiao, F., Jang, C. J., Zhang, L., and Song, Z.: Evaluation and projection of global marine heatwaves based on CMIP6 models, Deep Sea Research Part II: Topical Studies in Oceanography, 194, 104 998, https://doi.org/10.1016/j.dsr2.2021.104998, 2021.
- Padfar, S., Moftakhari, H., and Moradkhani, H.: Rapid intensification of tropical cyclones in the Gulf of Mexico is more likely during marine heatwaves, Communications Earth & Environment, 5, 421, https://doi.org/10.1038/s43247-024-01578-2, 2024.
  - Rieck, J. K., Böning, C. W., and Getzlaff, K.: The Nature of Eddy Kinetic Energy in the Labrador Sea: Different Types of Mesoscale Eddies, Their Temporal Variability, and Impact on Deep Convection, J. Phys. Oceanogr., 49, 2075–2094, https://doi.org/10.1175/JPO-D-18-0243.1, 2019.
- Roberts, J. M., Wheeler, A. J., and Freiwald, A.: Reefs of the Deep: The Biology and Geology of Cold-Water Coral Ecosystems, Science, 312, 543–547, https://doi.org/10.1126/science.1119861, publisher: American Association for the Advancement of Science, 2006.
  - Rühs, S., Oliver, E. C. J., Biastoch, A., Böning, C. W., Dowd, M., Getzlaff, K., Martin, T., and Myers, P. G.: Changing Spatial Patterns of Deep Convection in the Subpolar North Atlantic, Journal of Geophysical Research: Oceans, 126, e2021JC017 245, https://doi.org/10.1029/2021JC017245, publisher: John Wiley & Sons, Ltd, 2021.
- Scannell, H. A., Johnson, G. C., Thompson, L., Lyman, J. M., and Riser, S. C.: Subsurface Evolution and Persistence of Marine Heatwaves in the Northeast Pacific, Geophysical Research Letters, 47, e2020GL090 548, https://doi.org/10.1029/2020GL090548, publisher: John Wiley & Sons, Ltd, 2020.
  - Schaeffer, A. and Roughan, M.: Subsurface intensification of marine heatwaves off southeastern Australia: The role of stratification and local winds, Geophysical Research Letters, 44, 5025–5033, https://doi.org/10.1002/2017GL073714, publisher: John Wiley & Sons, Ltd, 2017.
- Schulzki, T., Henry, L.-A., Roberts, J. M., Rakka, M., Ross, S. W., and Biastoch, A.: Mesoscale ocean eddies determine dispersal and connectivity of corals at the RMS Titanic wreck site, Deep Sea Research Part I: Oceanographic Research Papers, 213, 104404, https://doi.org/10.1016/j.dsr.2024.104404, 2024.
  - Schulzki, T., Schwarzkopf, F. U., and Biastoch, A.: Atlantic wide detection of marine heatwaves beyond the surface in a high-resolution model [dataset], GEOMAR Helmholtz Centre for Ocean Research Kiel [distributor], https://hdl.handle.net/20.500.12085/49913d6b-4c70-43cb-9d3c-b4b73b0b8291, 2025a.

- Schulzki, T., Schwarzkopf, F. U., and Biastoch, A.: An Atlantic wide assessment of marine heatwaves beyond the surface in an eddyrich ocean model [dataset], GEOMAR Helmholtz Centre for Ocean Research Kiel [distributor], https://hdl.handle.net/20.500.12085/a3279a60-e9ef-437f-bd34-c3e156181e98, 2025b.
- Short, J., Foster, T., Falter, J., Kendrick, G. A., and McCulloch, M. T.: Crustose coralline algal growth, calcification and mortality following a marine heatwave in Western Australia, Continental Shelf Research, 106, 38–44, https://doi.org/10.1016/j.csr.2015.07.003, 2015.
- Smale, D. A., Wernberg, T., Oliver, E. C. J., Thomsen, M., Harvey, B. P., Straub, S. C., Burrows, M. T., Alexander, L. V., Benthuysen, J. A., Donat, M. G., Feng, M., Hobday, A. J., Holbrook, N. J., Perkins-Kirkpatrick, S. E., Scannell, H. A., Sen Gupta, A., Payne, B. L., and

- Moore, P. J.: Marine heatwaves threaten global biodiversity and the provision of ecosystem services, Nature Climate Change, 9, 306–312, https://doi.org/10.1038/s41558-019-0412-1, 2019.
- 955 Smith, K. E., Burrows, M. T., Hobday, A. J., King, N. G., Moore, P. J., Sen Gupta, A., Thomsen, M. S., Wernberg, T., and Smale, D. A.: Biological Impacts of Marine Heatwaves, Annual Review of Marine Science, 15, 119–145, https://doi.org/https://doi.org/10.1146/annurev-marine-032122-121437, 2023.
  - Smith, K. E., Sen Gupta, A., Amaya, D., Benthuysen, J. A., Burrows, M. T., Capotondi, A., Filbee-Dexter, K., Frölicher, T. L., Hobday, A. J., Holbrook, N. J., Malan, N., Moore, P. J., Oliver, E. C., Richaud, B., Salcedo-Castro, J., Smale, D. A., Thomsen, M., and Wern-
- berg, T.: Baseline matters: Challenges and implications of different marine heatwave baselines, Progress in Oceanography, 231, 103 404, https://doi.org/10.1016/i.pocean.2024.103404, 2025.
  - Stenvers, V. I., Hauss, H., Osborn, K. J., Neitzel, P., Merten, V., Scheer, S., Robison, B. H., Freitas, R., and Hoving, H. J. T.: Distribution, associations and role in the biological carbon pump of Pyrosoma atlanticum (Tunicata, Thaliacea) off Cabo Verde, NE Atlantic, Scientific Reports, 11, 9231, https://doi.org/10.1038/s41598-021-88208-5, 2021.
- Sun, D., Li, F., Jing, Z., Hu, S., and Zhang, B.: Frequent marine heatwaves hidden below the surface of the global ocean, Nature Geoscience, 16, 1099–1104, https://doi.org/10.1038/s41561-023-01325-w, 2023.
  - Tsujino, H., Urakawa, S., Nakano, H., Small, R. J., Kim, W. M., Yeager, S. G., Danabasoglu, G., Suzuki, T., Bamber, J. L., Bentsen, M., Böning, C. W., Bozec, A., Chassignet, E. P., Curchitser, E., Boeira Dias, F., Durack, P. J., Griffies, S. M., Harada, Y., Ilicak, M., Josey, S. A., Kobayashi, C., Kobayashi, S., Komuro, Y., Large, W. G., Le Sommer, J., Marsland, S. J., Masina, S., Scheinert, M., Tomita, H.,
- Valdivieso, M., and Yamazaki, D.: JRA-55 based surface dataset for driving ocean–sea-ice models (JRA55-do), Ocean Modelling, 130, 79–139, https://doi.org/10.1016/j.ocemod.2018.07.002, 2018.
  - Tsujino, H., Urakawa, L. S., Griffies, S. M., Danabasoglu, G., Adcroft, A. J., Amaral, A. E., Arsouze, T., Bentsen, M., Bernardello, R., Böning, C. W., Bozec, A., Chassignet, E. P., Danilov, S., Dussin, R., Exarchou, E., Fogli, P. G., Fox-Kemper, B., Guo, C., Ilicak, M., Iovino, D., Kim, W. M., Koldunov, N., Lapin, V., Li, Y., Lin, P., Lindsay, K., Liu, H., Long, M. C., Komuro, Y., Marsland, S. J., Masina, S.,
- Nummelin, A., Rieck, J. K., Ruprich-Robert, Y., Scheinert, M., Sicardi, V., Sidorenko, D., Suzuki, T., Tatebe, H., Wang, Q., Yeager, S. G., and Yu, Z.: Evaluation of global ocean–sea-ice model simulations based on the experimental protocols of the Ocean Model Intercomparison Project phase 2 (OMIP-2), Geosci. Model Dev., 13, 3643–3708, https://doi.org/10.5194/gmd-13-3643-2020, publisher: Copernicus Publications, 2020.
- Vinha, B., Murillo, F. J., Schumacher, M., Hansteen, T. H., Schwarzkopf, F. U., Biastoch, A., Kenchington, E., Piraino, S., Orejas, C., and Huvenne, V. A. I.: Ensemble modelling to predict the distribution of vulnerable marine ecosystems indicator taxa on data-limited seamounts of Cabo Verde (NW Africa), Diversity and Distributions, 30, e13 896, https://doi.org/10.1111/ddi.13896, publisher: John Wiley & Sons, Ltd, 2024.
  - WMO-No.1203: WMO Guidelines on the Calculation of Climate Normals, https://community.wmo.int/en/wmo-climatological-normals, 2017.
- Wu, T. and He, R.: Gulf Stream mesoscale variabilities drive bottom marine heatwaves in Northwest Atlantic continental margin methane seeps, Communications Earth & Environment, 5, 574, https://doi.org/10.1038/s43247-024-01742-8, 2024.
  - Wyatt, A. S. J., Leichter, J. J., Washburn, L., Kui, L., Edmunds, P. J., and Burgess, S. C.: Hidden heatwaves and severe coral bleaching linked to mesoscale eddies and thermocline dynamics, Nature Communications, 14, 25, https://doi.org/10.1038/s41467-022-35550-5, 2023.

- Xu, T., Newman, M., Capotondi, A., Stevenson, S., Di Lorenzo, E., and Alexander, M. A.: An increase in marine heatwaves without significant changes in surface ocean temperature variability, Nature Communications, 13, 7396, https://doi.org/10.1038/s41467-022-34934-x, 2022.
  - Zeng, Y. and Janjić, T.: Study of conservation laws with the Local Ensemble Transform Kalman Filter, Quarterly Journal of the Royal Meteorological Society, 142, 2359–2372, https://doi.org/10.1002/qj.2829, publisher: John Wiley & Sons, Ltd, 2016.
- Zenk, W., Klein, B., and Schroder, M.: Cape Verde Frontal Zone, Deep Sea Research Part A. Oceanographic Research Papers, 38, S505–S530, https://doi.org/10.1016/S0198-0149(12)80022-7, 1991.
  - Zhang, Y., Feng, M., Du, Y., Phillips, H. E., Bindoff, N. L., and McPhaden, M. J.: Strengthened Indonesian Throughflow Drives Decadal Warming in the Southern Indian Ocean, Geophysical Research Letters, 45, 6167–6175, https://doi.org/10.1029/2018GL078265, publisher: John Wiley & Sons, Ltd, 2018.
- Zhang, Y., Du, Y., Feng, M., and Hobday, A. J.: Vertical structures of marine heatwaves, Nature Communications, 14, 6483, https://doi.org/10.1038/s41467-023-42219-0, 2023.
  - Zweng, M. M., Reagan, J. R., Antonov, J. I., Locarnini, R. A., Mishonov, A. V., Boyer, T. P., Garcia, H. E., Baranova, O. K., Johnson, D. R., Seidov, 1948, D., Biddle, M. M., and Levitus, S.: World ocean atlas 2013. Volume 2, Salinity, https://doi.org/10.7289/V5251G4D, 2013.

Figure 10. Fraction of the area in the Cape Verde archipelago that is occupied by a MHW on any given day in the 6th cycle of VIKING20X when applying the WMO ocean heat content anomaly (relative to the 1980-2009 daily climatology; a) and linear detrended ocean heat content anomaly from the daily climatology (b)baselines. Values smaller than 10% are not shaded. The black line indicates the maximum mixed layer depth within the region. MHW events detected in the 6th cycle of VIKING20X with the fixed 30yr. (a) and detrended (b) baselines are shaded grey. Vertical profiles of the contribution of different heat budget terms to individual MHW eoverage variance explained by its values in 5 selected depths events detected with the fixed 30yr (c<sub>7</sub>) and detrended (d) baselines. The contributions were averaged over all MHWs that occur at a particular depth. OHT - horizontal ocean heat transport, OHT<sub>w</sub> - vertical ocean heat transport, Res - residual flux. Diamonds indicate the surface heat flux contribution in first model level.

Figure 11. Ocean Vertical structure of the heat content anomaly budget in the Cape Verde archipelagoand 15% MHW coverage contour based on the WMO baseline. Horizontal (VIKING20X-6th; a) . Day to day ocean heat content change and 15% MHW coverage contour based on the linear baseline vertical (VIKING20X-6th; b) . Vertical profiles of the mean (1980-2022) heat fluxes from horizontal/vertical ocean advection (OHT/Vert), the surface heat flux (HFX) and the residual (Res) that is mostly related transport anomalies relative to vertical mixingthe 1980-2009 daily climatology. The resulting mean ocean heat content change (△OHC) is shown in orange and its values are indicated. To emphasise sustained anomalies, that have a strong impact on the top x-axis (e). Fraction of the heat contentehange variance explained by different heat budget terms for all depth levels in the model (d). A 10-day running mean was applied to remove, high frequency variability was removed by applying a 60-day moving average. Normalised magnitude of the horizontal temperature gradient Mean (VT1980-2022) difference between the domain average temperature and speed across the boundaries temperature along the boundary of the Cape Verde archipelago and average vertical temperature gradient region (dT/dzc)and vertical velocity. Mean (W1980-2022) within the region vertical temperature gradient. The inlet shows Note the same, but with different changing x-axis limits caling for the top 300 m in c) and d).

**Figure A1.** Differences between the coarser resolution ORCA025 and high-resolution VIKING20X configurations. Mean kinetic energy (a), eddy kinetic energy (b), eddy kinetic energy of the vertical velocity (c), vertical temperature gradient (d) and Horizontal temperature gradient (e). All maps show the difference between the mean quantities (1980-2022). The temperature trend (linear regression slope) difference is based on the same time period.