# Peer review of "An Atlantic wide assessment of marine heatwaves beyond the surface in an eddy-rich ocean model"

_EGUsphere, 2025_

## Referee Comment (RC2)

**Review**: An Atlantic wide assessment of marine heatwaves beyond the surface in an eddy-rich ocean model

This study uses two model configurations with eddy permitting and eddy-rich horizontal resolution to assess the influence of the representation of mesoscale dynamics on surface and deep MHWs in the Atlantic ocean. The effect of the model drift and the baseline, against which MHWs are detected, are investigated with respect to the events' characteristics. Finally the study examines the physical processes that seem to control the heat content in the Cape Archipelago, in an attempt to explain what controls the development of MHWs in that region. Although I find this study useful and quite novel, in some aspects, for the MHW community, the current structure and writing of the paper does not help the reader to follow key results with a coherent flow and meaning. Given the considerable length of the paper and the often unstructured arguments and experiment description the reader is often lost on where they should look and focus. Therefore, although I believe this study has a potential for publication, it needs a major re-structuring and re-writing of its content. Below I provide detailed major and minor comments that could help the writers improve the clarity of their paper. I advice for publication but only after major revision is performed.

Major Comments:
**1**. Although generally well-written, the paper could profit from a native English speaker to further improve the language and syntax throughout the manuscript. This would improve the clarity of the paper for the benefit of the reader.

**2**. The heat budget formulation used in the paper, aims at representing advection fluxes throughout partial interfaces of the selected region (Equations 2 & 3). However, the way that is currently mathematically formulated is rather ambiguous, according to Lee et al., (2004) and Kim et al., (2006), mostly representing internal processes redistributing heat within the domain under investigation. Unless this is the overall goal of the writers, I would advise reading those two works and reformulating the mixed layer heat budget to represent the external processes that control the domain's heat content and then estimate the effect of the heat contribution through each of the region's boundaries, following the proposed mathematical formulation. Alternatively, the writers can use the traditional mathematical formulation of the heat budget, but assessing the heat contribution of the to total (horizontal and vertical) advection instead.

References:
Lee, T., Fukumori, I., & Tang, B. (2004). Temperature advection: Internal versus external processes. *Journal of Physical Oceanography*, *34*(8), 1936-1944.
Kim, S. B., Fukumori, I., & Lee, T. (2006). The closure of the ocean mixed layer temperature budget using level-coordinate model fields. *Journal of Atmospheric and Oceanic Technology*,*23*(6), 840-853.

**3**. The description of the results is often ambiguous due to the lack of clear references to the corresponding figures, which may lead to reader confusion. I strongly recommend that the authors improve the numbering of their figures and carefully review the manuscript to ensure that each figure and its subplots are cited appropriately within the relevant text. Please refer to my comments below for specific suggestions.

**4**. Some methodological choices in the study, such as the selection of the 1st and 6th cycle of the simulation, are not clearly justified at the beginning of the paper but are introduced later. To enhance clarity and avoid potential confusion for the reader, I recommend that the authors provide a clear and explicit justification for all methodological decisions early in the manuscript. Please refer to my comments below for specific suggestions.

**5**. The description of the results is often intermingled with elements that belong in the discussion, which detracts from the clarity of the paper. I recommend that the authors more clearly separate these two sections, while also enhancing the structure and clarity of the methods section.

Minor – Detailed Comments:

**Line 104**: Could you clarify the specific rationale behind selecting the 1st and 6th cycle of the model for analysis? This section is where you should define the key characteristics of the simulations and justify your choices, ensuring that readers understand the reasoning behind them. Providing this information early on will help familiarize readers with the terminology and framework, allowing for a clearer interpretation of the results later in the paper.

**Lines 125 – 130**: I recommend that the authors present a more structured and clearly organized overview of the different simulations/experiments conducted, along with their key characteristics. This could be achieved through a matrix or bullet points, explicitly highlighting the distinctions between the two experiments. A clearer presentation of these details will enhance the readability of the manuscript and provide readers with a solid understanding of the analysis that will follow.

**Line 134**: Does that mean that the results will be the same with other baseline approaches (e.g. Smith et al., 2025)?Clarify.

**Line 172- Equation 2**: I understand that this is the conventional approach to formulating horizontal advection flux through each boundary of a given area or section. However, as demonstrated by Lee et al. (2004) and Kim et al. (2006), this formulation renders the conventional temperature flux through a partial boundary problematic. Their research highlights that temperature flux, when expressed in this manner, becomes ambiguous because mass is typically not conserved through a partial interface. Consequently, the advection flux, as derived from the Gauss/divergence theorem, cannot be meaningfully decomposed into western, eastern, southern, and other directional components as presented here. Instead, they propose an alternative formulation that ensures the temperature flux through a partial section remains meaningful by referencing it to the domain-averaged temperature. I advise the writer to carefully read the studies and modify their calculations according to what they would like to represent (total advection flux or advection through a partial boundary)

**Line 180**: Which depths does the integration happen?This needs to be clarified from the beginning to provide the readers with a solid understanding of the analysis that will follow.

**Lines 200-202**: Which figure do these results correspond to? Please provide a clear reference to the corresponding figures.

**Line 209**: Do you mean Figure 3a-d? The figure subplots should have a different numbering/letter in order to tell them apart more easily.

**Lines 210-214**: Please, provide a clear reference of the results described here to the corresponding figures.

**Lines 214-215**: Could you clarify further the connection between the MHW threshold behavior at 2200m and the mid-depth ocean warming observed in the 1980s compared to the 1990s? The relationship between these two aspects is not immediately clear.

**Line 217**: To ensure clarity and consistency for the reader, it is best to use a single term throughout the text. Either "detrending" or "linear increase in baseline" or even better find more intuitive names for the experiments and use throughout the text. This avoids potential confusion

**Lines 218-219**: "*Therefore, it is not possible to recover the MHW statistics from non-detrended temperatures in the 6th cycle by detrending the 1st cycle (e.g. 1st-linear does not match 6th-fixed/WMO)*". Could you clarify whether you mean that one should not expect similar MHW statistics when comparing non-detrended temperatures (which include both climate trends and model drift) to temperatures detrended using only the first cycle? What would be the rationale for such an approach? In the first case, strong model drift is present, whereas in the second, detrending removes real-world climate trends. It is unclear why one would expect comparable results under these conditions. Additionally, the distinction between the first

and sixth cycle temperatures should be more rigorously defined in the Methods section using precise scientific terminology

**Line 222**: Line 222: Did you mean Figure 3a, b? Additionally, the red lines in Figure 3a and 3b appear similar, could you clarify where the dissimilarities are?

**Lines 224-225**: Problem with the syntax of this sentence.

**Lines 226-227**: What do you mean disadvantages due to finite length of the timeseries. Could you elaborate this more so that the reader understands it better?

**Figure 3d**: The two regions should be highlighted in different colors and explicitly referenced in the text for clarity. Additionally, why are only the results of the WMO shown, while those of the linear 1st and linear 6th are omitted? Also, it would be helpful to include a brief description of what the 1st and 6th cycle represent in the caption of the figure, for the benefit of the reader.

**Line 229**: Could you clarify how the MHW statistics were calculated? Were events identified separately at each grid point and depth level across the entire Atlantic, followed by averaging over longitude and latitude? Or was the Atlantic first averaged at each latitude and longitude, reducing the dataset to only the vertical dimension, with the MHW detection algorithm then applied separately at each level? Or were MHWs first identified at the surface, with MHW days at depth determined based on the timing of surface MHW events? Please clarify the methodology used for constructing the vertical MHW dataset, preferably in methods section.

Additionally, what explains the minimal differences between the 1st linear and 6th linear experiments? Why are the differences in the number of MHW days between the first and last 10 years so small?Clarify. Also what about the differences between the 1st and 6th linear in FIgure 3d?why are they not mentioned anywhere?

**Lines 236-240**: This section leans more toward discussion rather than results, which should focus solely on reporting numbers, percentages, and observed changes. Any further interpretation or commentary should be reserved for the discussion section.

**Line 241**: m" in any case" is not an appropriate scientific expression. Rephrase

**Lines 242-248**: This section leand more towards conclusion than results.

**Lines 246 - 247**: What is the rationale behind selecting this specific type of simulation for your analysis? A clear explanation and objective justification for this choice are necessary, either early in the methodology, where all the steps of your methods should be clearly defined, or here.

**Line 252:** Why did you select maximum intensity as a metric for comparison instead of mean intensity? A clear and well-supported justification is needed. Also, while the model generally underestimates maximum intensity across most of the Atlantic, it appears to overestimate it in the Gulf Stream region. Therefore, this argument is not exactly correct.

**Line 254**: What do these statistics represent?an average over the Atlantic? This should be mentioned in the text. not only in the figure caption.

**Lines 263-265:** How can model and observations agree on regions with longer and shorter durations but at the same time the differences between high and low durations be more pronounced in the model?That sentence is confusing. Clarify

**Line 268**: How does the vertical resolution of the model play a role when talking about differences in surface MHW characteristics?Unless the writers mean something else. Please clarify.

**Line 270**: The term "broken section" is not scientific. Please, rephrase.

Also, it is unclear how the black lines in Figure 5 correspond to the regions mentioned. Are they representing a simple vertical section along specific points across the Atlantic? If so, the exact latitude and longitude coordinates defining the start and end of these sections should be clearly documented in a table rather than solely in the figure maps, ensuring reproducibility. Alternatively, do these lines represent a spatially averaged area around the sections? This distinction needs to be clarified. Furthermore, the rationale behind selecting these specific regions should be explicitly stated. What criteria were used to define these areas, and why were they chosen for analysis? A clear justification is necessary.

**Lined 271 - 272**: Syntax error in the sentence.

**Line 275**: Which area does this sentence refer to?

**Lines 277–278**: It is unclear whether this statement is directly related to the preceding sentence or introduces a separate result. Additionally, the specific section to which this sentence refers is not clearly defined. Clarification is needed to ensure coherence and precision.

**Line 290**: does Labrador Sea correspond to Canada section in FIgure 5? Not clear. It is better to keep a consistent name of your areas both in the text and in the figures.

**Line 295**: For clarity and to benefit the reader, it is essential to use the same regional names in the text as those used in the figure. Additionally, please refer to the corresponding subplot in Figure 5 to further enhance clarity.

**Line 300-302**: It is important to reference specific subplots of Figure 5 and use consistent terminology for the regions throughout the text. Additionally, the explanation provided here appears to be more suited for the discussion section. Please ensure that the results section remains focused strictly on presenting the findings, without further interpretation.Also the MLD does not increase. Instead it gets shallower.

**Figure 5**: The subplots of this figure need to be named separately for clarity purposes and ease of reading in the text.

Lines 212-213: Can you provide evidence to support this, or is this a claim based on theoretical assumptions? It is important to substantiate this statement with relevant data, references, or a clear rationale to strengthen the argument.

**Lines 316,320,323**: You need to reference the relevant figure here, as it is unclear which one the reader should be looking at. Including a specific figure reference will enhance clarity and guide the reader effectively.

**Lines 317-318**: I do not understand this sentence. Do you mean that the interior and near coastal regions of the ocean, have similar sea-floor, depths? This sounds strange. Also, what does it mean" *even if the sea floor is located in similar depths*"? What would happen if the sea-floor was located in different depths (which it does not, at least comparing the coastal and interior areas of the ocean).

**Line 326**: It is unclear what is meant by "linear trend" in this context. Does it refer to a trend in temperatures or a trend applied to identified MHWs? Please clarify the specific variable or process being referenced by the term "linear trend."

**Lines 327-329**: These sentences require improved grammatical flow and syntax, in addition to clear references to the appropriate subplots of Figure 6 for enhanced clarity. What specific trends are being referred to? Are these linear trends in temperature, or do they pertain to trends in MHW duration or intensity? The current phrasing is unclear and needs further specification.

**Lines 339 - 340**: Please provide a justification for your choice of the WMO baseline over the linear one. Additionally, clarify where you demonstrate that the conclusions are independent of the baseline selection.

**Lines 348-349**: The statement regarding the mean frequency being similar, yet higher in VIKING20X, is contradictory. Please clarify how these two observations can be reconciled.

**Lines 350-351**: These sentences belong to discussion

**Lines 355-357**: In addition to the fact that this sentence belongs in the discussion section, how can the entire Northeast Atlantic MHW features be attributed solely to the Mediterranean outflow, rather than interactions between the North Atlantic Current and waters of (sub)polar origin? Please provide evidence to support this claim.

**Lines 358-370**: These sentences belong to the discussion.

**Lines 376**: Have you shown that somewhere?supplementary material perhaps?not clear.

**Line 391**: The phrase "are not in phase" is not the most appropriate syntax here. An alternative phrasing could be: "The surface MHW characteristics differ from those observed at depth." Additionally, you need to specify which subplot of Figure 9 you are referring to, as it is currently unclear where the reader should focus.

**Line 392**: Could you clarify what you mean by "explained variance"? Was a statistical test conducted to analyze the MHW characteristics?

**Lines 393-394**: How do the atmospheric heat fluxes relate to this figure, which focuses on MHW characteristics?

**Lines 396-397**: The decoupling of surface and deeper MHWs is generally not attributed to long-term trends. Instead, the distinct variability and decoupling of MHW characteristics are typically the result of differing oceanic processes at the surface (high-frequency variability) and at depth (low-frequency variability). Therefore, the physical relevance of this sentence is unclear.

**Line 400**: Again, what is explained variance here? Was there a statistical test performed here?

**Lines 403-404:** If this refers to a different dataset or data handling process, it is essential to provide a clear reference to the methods section for better context and understanding. Please ensure that the relevant details are adequately explained and linked to the methodology for clarity.

**Line 406:** Please ensure that the subplots in Figure 5 are clearly labeled and referenced in the text. This will help guide the reader to the correct location for the relevant information. The current lack of clarity makes it difficult to understand which specific subplot is being referred to.

**Lines 406-407**: The typical range of MHW variability is unclear in this context. Please ensure that you reference the appropriate figure or subplot to guide the reader. Without this reference, it is difficult to determine which data you are referring to.

**Line 411**: The statement that MHWs show no considerable trends at this depth appears contradictory. From Figures 9a-c, it is evident that there is a higher number of events at the beginning of the timeseries, with a decreasing trend toward the end. Please clarify this inconsistency.

**Lines 412-413**: These sentences belongs to methods. Also the phrase "MHWs vary in phase" everywhere in the text. This sentence is unclear and contains syntax errors. It is important to specify whether this refers to the relationship between surface and subsurface layers or if it applies to other layers. Please clarify the intended meaning, or consider rephrasing with a more precise expression for better clarity.

**Figure 9**: Depth ranges between Figures a-c is up to 3800m while in Figures d-f up to 1500m. This inconsistency should be addressed. Consistency across the figures is essential for clarity and accuracy in presenting the data.Also in the caption of this figure, a new simulation is mentioned "*WMO baseline but with the trend removed after calculating the annual mean MHW characteristics*". However, this simulation/experiment has not been mentioned previously in the Methods section, making it the first reference to it in the figure caption. It would be more appropriate to introduce and explain this simulation in the Methods section to adequately prepare the reader before mentioning it in the figure caption.

**Line 420**: It is not clear which specific subplot of Figure 10 is being referenced here, nor which experiment these results pertain to. It is essential to clearly specify the relevant subplot and provide context regarding the experiment to avoid confusion for the reader.\

**Line 422**: This sentence is unclear. What is meant by stating that the linear trend had a minor impact on the Cape Verde Archipelago? The linear trend can only affect the MHW characteristics in the Cape Verde region, not the region itself. Please clarify.

**Line 423**: The MHW coverage differs between the two simulations (Fig. 10a-b), particularly at depth, and is not similar at the surface.Clarify.

**Lines 425-426**: I do not understand how can you physically connect temperature anomalies (that are above the 90th percentile) and that have been identified at each layer separately. The application of the Hobday et al., 2016 at each layer does not guarantee spatial coherence of events at surface or at depth and so the events are not demonstrated to be connected from one layer to the other just because there is a temporal continuation of temperature anomalies. How can you know if the process that is responsible for the development of temperature anomaly at let's say 600m is physically and mechanistically related to the temperature anomaly identified at surface?This is an implicit assumption which can be made only by looking at the figures but it is not physically proven. Especially because these are anomalies above the 90th percentile at each layer. If an anomaly at a selected depth is below the90th percentile for some days but then comes back above the threshold, will it still be considered as a MHW?This is a problem inherent to the statistical MHW framework. Therefore assumptions of the vertical continuation of MHWs should not be made lightly.

**Line 430-434**: You need to reference the appropriate subplot of the figure for clarity. It is currently unclear which part of the figure these sentences refer to.

**Lines 437**: The specific figure being referenced here is unclear, and the concept of "disconnection between the two depth layers" is not well-explained.

**Lines 438- 439**: Please provide a clearer explanation of this result, and ensure the appropriate figure is cited for better clarity.

**Line 441-442**: Has this long-term cooling trend seen anywhere?An appropriate figure (subplot) should be cited here.Otherwise this looks as a general statement without proof.

**Lines 443-444**: The similarity being discussed is between the simulations/experiments that identify MHWs using the WMO and linear baselines, not between the baselines themselves. This distinction needs to be clearly defined throughout the text. As it stands, the sentence is unclear and requires better clarification.

**Lines 446- 455**: This paragraph belongs to the discussion.

**Lines 457-458**: The phrase "hint on connection to the surface" is unclear. To improve clarity, the authors should specify what indicators or evidence suggest a connection between surface and subsurface events

**Line 464**: The base of the mixed layer should be clearly defined in the text or figure legend, as it is not evident from the figures. Additionally, it is essential to reference the specific figure or subplot to ensure the reader knows exactly where to look for the relevant information. For clarity, the layer's depth or definition should be described in relation to the figure presented.

**Line 465**: As previously mentioned, the MLHB format utilized in this study introduces ambiguity when representing temperature flux across a partial interface (x, y, z). It currently reflects the redistribution of heat within the domain rather than the change in heat content at the interface due to temperature variations. See my comment above

**Line 468-472**: This description probably refers to Figure 1? however, as no specific figure is cited, it is unclear where these results correspond to. Please clarify by referencing the appropriate figure (subplot) to ensure the reader knows where to look. Currently, this sentence is vague.

**Line 474 - 475**: This sentence is unclear. The sentence cites Figure 11b, discussing the impact of removing the linear trend, while the caption of Figure 11b describes the ocean heat content change based on the linear baseline. Please clarify the relationship between the two, as the current wording is contradictory.

**Lines 475-476**:Have you shown or read that before?Otherwise it is an arbitrary statement.

**Lines 477-478**: This sentence is confusing. Please clarify or rephrase.

**Lines 478 - 483:** Once again, the specific location of these results is unclear, and it is not evident which figures to refer to. Additionally, why is the description of the heat content provided for only one experiment/cycle used in the study, and not for both? Is there a notable difference between them? Please clarify.

**Line 484**: This statement requires proper citation of figures. Currently, it is unclear which figure is being referenced. Please ensure that the relevant figures are cited explicitly for clarity.

**Lines 486-487**: Could you please clarify the depth of the mixed layer base? Additionally, the colors used in Figure 11e are not distinct enough to clearly distinguish the different processes described. I recommend adjusting the color scheme for better clarity and differentiation of the processes.

**Line 497**: Could you clarify what you mean by "changing relative importance"? The expression is unclear. A rephrasing or further explanation would help ensure clarity.

**Lines 501-511**: This paragraph is more conclusion section and not results.

Figure 11: Could you specify the climatology period used to calculate the ocean heat content anomaly plot? This is currently unclear.

**Line 546**: Could you clarify what you mean by stating that the WMO and fixed baselines are not applicable below 100m? What specific application or context are you referring to?

**Line 565**: Could you clarify what you mean by "in the upper ocean, temperatures cover a large range"? Temperature is a fundamental and ubiquitous property of the ocean, so it would be helpful to specify what aspect of the temperature variability or range you are referring to.

**Lines 585-586**: Once again, I do not believe the influence of the Mediterranean Sea Outflow (MSO) has been adequately demonstrated here. It appears to be inferred based on unclear deductions. Please provide compelling evidence to support the claim that the variability described is indeed related to the MSO, or alternatively, clearly state it as a speculative observation. Additionally, which specific depth are you referring to? Given that the study examines multiple depths throughout the water column, this statement may not hold true across all depths analyzed. Please be more precise in your reference to the depth in question.

**Line 588 - 591**: For which of the experiments conducted here does this apply? Or are these observations valid for both experiments? Please clarify.

**Line 593**: A decrease in extreme positive temperatures? How is this observed? Which figures illustrate this? Please clarify.

**Lines 614-615**: Which publication. give examples.

**Line 619**: The word "pitfall" is not an appropriate word for a scientific paper. Use, limitation or challenge instead.

---

## Community Comment (CC2)

**Literature Review**

[Figure]

**Berra Lorenzo**

Corso di Laurea in Fisica, course in Physics of the Hydrosphere and the Cryosphere. Report on paper: https://egusphere.copernicus.org/preprints/2025/egusphere-2025-571/, An Atlantic wide assessment of marine heatwaves beyond the surface in an eddy-rich ocean model

**1 Modeling**

Forcing conditions are applied to the model and presumably include heat, currents and air circulation, which are all mentioned in the study without explicitly reporting the full scope of said conditions.

Simulations are said to be performed using the VIKING20X following the OMIP-II protocol, prescribing six consecutive simulations spanning the 1958 to 2019 time-frame, with the first one initialising from WOA13 data and oceanic conditions at rest, while each of the following cycles are initialised using the final oceanic conditions of the previous cycle; each cycle is then extended up to 2023 to analyse the 1980 to 2022 time-frame. In particular, only the first and sixth cycle of each series are analysed: I infer this is done to observe an immediate response to the forcing conditions in the first cycle and the influence of model drift and model spin-up in the last cycle. While I believe this is needed due to the limited timeframe in which data is available I could not find evidence of the validity of this cycle-based approach for the simulations either in this paper or in the provided reference (Tsujino et al., 2020), as after a new cycle has started oceanic conditions at a certain time $t$ in the time-frame provided (1958-2019) would instead be mapped in the simulation to a time $t' = t + (n-1)\Delta$, where $n$ is the number of the cycle being computed and $\Delta$ is the length of the time-frame, to my understanding.

**2 Results**

The influence of geothermal activity on MHWs is not mentioned in the paper, and as such I would like to inquire if it is speculated to be noticeable, especially on bottom MHWs, or would stable geothermal activity not impact MHW formation due to their statistical definition?

The effects of model drift are shown to be greatly reduced in the linearly-increasing baseline. Being model drift defined as the adjustment of the simulated environment to unknown initial conditions, could it be argued that the primary effect of these unknown initial conditions is the temperature rise, thus reducing the model spin-up time needed, and that other lesser effects of said conditions are higher-order corrections?

The simulations performed with this new protocol are said to not be decidedly more realistic than previous ones. Since the main difference from previous models is the impact of mesoscale dynamics, which could either have a cumulative impact or be averaged out over larger portions of the ocean, could these lower scale dynamics be seen as higher-order terms in the model approximations? If so a convergence interval should be defined, where the model can be argued to be more realistic, while outside of it higher-order corrections may not yield better approximations.

Being MHWs defined as events lasting at least five days, and since MHWs divided by less than two days are considered the same MHW, would this merging of MHWs cause problems for the MHW frequency data in areas where they tend to have longer durations along the length of the simulations?

The heat budget present in this study doesn't contain, at least from what is shown, the influence of the night-day cycle. Since MHWs at shallow depths are shown to be highly responsive to external conditions it could be an interesting forcing condition, but it may

very well be averaged out over the multi-decadal time-scales used for the simulations, and it would outgrow the focus of the study on greater depth MHWs.

---

## Author Comment (AC1)

**Response to reviewers**

**Review RC1**

This paper is a nice analysis of marine heatwaves with depth from an ocean model, which emphasises the role of model drift, baseline choice and resolution on marine heatwaves detection and statistics. Vertical coherence and drivers are also discussed as part of a case study.

The paper is interesting, and a good contribution to the field. However, I think that recommendations around baselines and spin up periods could be clarified, and believe that the heat budget analysis, while not incorrect, may not be fit for purpose.

Thank you for your positive feedback and helpful suggestions to improve our manuscript. As suggested, the revised manuscript contains additional and more clear recommendations and an updated heat budget analysis. Please find a detailed response to the review comments below.

**Major Comments**

Heat budget

I have concerns about the application of the heat budget. As used in the paper, it allows the diagnosis of the drivers of changes in heat content. However, MHWs are defined as discrete threshold exceedances relative to a local climatology, not by absolute temperature tendencies. Thus, areas of persistent heat convergence are not necessarily directly comparable to discrete MHW events. The manuscript currently suggests a causal link between sustained heat convergence and MHW occurrence or vertical coherence, but does not address this fundamental distinction. In order to prove this causal link, I believe that the heat budget would have to be performed on an event-by-event basis, and vertical coherence would have to be considered very carefully in terms of the boundary conditions for the budget for each event.

While this would be a very interesting analysis, I think it would be beyond the scope of the paper in its current form. In fact, I do not think that the heat budget analysis adds much to the outcomes of this work, and so my recommendation would be to restrict the analysis to that of vertical coherence, and remove, or at least strongly tone down and place the heat budget in the context of warming, and not of MHWs.

Thank you for this suggestion. We fully agree that the link between individual MHWs and the heat budget analysis was too vague. As a consequence, we have replaced these results by an event based approach, as suggested. For that we detected MHWs based on the spatially averaged temperature of the Cape Verde archipelago. In contrast to the area covered by MHWs this allows to obtain a well defined start and end date of individual MHWs events. We then calculate the contribution of the different heat budget terms to the detected MHWs for

each depth level to identify the dominant drivers of MHWs (new figure 10). In a last step we explain how and why the main drivers of individual events are different in certain depth ranges (new figure 11). Corresponding changes can be found in the method section (2.4) and section 3.4

**Recommendations**

The paper makes recommendations around the use of sufficient model spin up periods, as well as the resolution at which MHW statistics should be calculated. I think that these recommendations will be very useful to researchers planning experiments for MHW use. However, I think the recommendations could be clarified in the text as they are sometimes not clearly laid out.

We thank you for this suggestion and extended our discussion of spin-up strategies and the resolution in the discussion section. We have added more specific recommendations although for the spin-up strategy there is no universally applicable strategy (see our response to your comment below).

Is it possible to include more detail about the spin up process required? How much drift is removed in, for example the 4$^{th}$ cycle as opposed to the 6$^{th}$? Would a repeat year forcing spin up be sufficient?

We chose to use the two extreme cases here, the shortest and longest spin-ups available for the 1980-2022 time period. The exact spin-up time needed depends on depth, model configuration and forcing. For example, the drift in mid-depth water masses is already small in the 3rd cycle, but as mentioned in the manuscript, the bottom waters may not be in equilibrium even in the 6th cycle. A repeat year spin-up could be sufficient and would allow for distinguishing between intrinsic model drift and forced trends. Since we have no experiment with a repeat year forcing any in-depth comment about this would however be pure speculation. As a consequence, we can only conclude that an adequate spin-up is needed. It should be monitored whether the deep water mass properties stabilise over the course of the spin-up, but we can not provide a specific time or procedure that is valid beyond the procedure we have tested in our VIKING20X configuration here. Nevertheless, we have added more recommendations regarding the spin-up required at different depths in the discussion section (lines 578-597).

I find the finding about resolution very compelling, i.e. that while high resolution is needed to resolve mesoscale processes, MHW statistics can, in most cases be calculated on a coarser resolution grid. I note that this finding is not emphasised in either the abstract or conclusions of the paper and would suggest that it should be.

We thank you for your positive feedback on our study. We have added this result to the abstract. We think it is already mentioned in the Summary and Conclusion section, but we have put more emphasis on this aspect (lines 539-567).

**Minor Comments**

I found the terminology and methology related baselines a little confusing at times. Smith et al (2025) recently published a detailed investigation of the effects of different baselines. For consistency with future literature, I would suggest citing this paper, and adopting their terminology (e.g. 'detrended baseline' instead of 'linear baseline')

Thank you for suggesting this study. We have changed the terminology to follow Smith et al. (2025) throughout the manuscript.

I'm confused about the difference between model drift and real temperature trends – using a detrended baseline will remove both indiscriminately, while using a long enough spin up will remove model drift, but preserve natural temperature trends – is this correct? If so, it should be made clearer in the manuscript.

Yes, this is correct. There are "real" trends that are for example caused by the surface forcing (warming, increase in wind stress, …). Additionally, there are trends that only arise from the model adjusting to the initial state, which is not the model's equilibrium state. With a sufficiently long spin-up the forcing related trends are still simulated, but the model is closer to its equilibrium state and therefore model drift is reduced.

**References**

Smith, K. E., Gupta, A. S., Amaya, D., Benthuysen, J. A., Burrows, M. T., Capotondi, A., ... & Wernberg, T. (2025). Baseline matters: Challenges and implications of different marine heatwave baselines. *Progress in Oceanography*, *231*, 103404.

---

## Author Comment (AC2)

**Response to reviewers**

**Review RC2**

We thank you for evaluating our manuscript and providing very detailed suggestions to enhance the clarity and readability of our manuscript. As suggested, we have changed the definition of the heat budget, made a clearer separation between results and discussion sections and provided several clarifications throughout the manuscript, including more references to specific figures (panels). Please find a detailed response to all review comments below.

**Major Comments:**

1. Although generally well-written, the paper could profit from a native English speaker to further improve the language and syntax throughout the manuscript. This would improve the clarity of the paper for the benefit of the reader.

We thank you for the feedback and have done our best to improve the structure and readability of the manuscript.

2. The heat budget formulation used in the paper, aims at representing advection fluxes throughout partial interfaces of the selected region (Equations 2 & 3). However, the way that is currently mathematically formulated is rather ambiguous, according to Lee et al., (2004) and Kim et al., (2006), mostly representing internal processes redistributing heat within the domain under investigation. Unless this is the overall goal of the writers, I would advise reading those two works and reformulating the mixed layer heat budget to represent the external processes that control the domain's heat content and then estimate the effect of the heat contribution through each of the region's boundaries, following the proposed mathematical formulation. Alternatively, the writers can use the traditional mathematical formulation of the heat budget, but assessing the heat contribution of the to total (horizontal and vertical) advection instead.

Thank you for pointing us to this better formulation of the heat budget. When calculating this heat budget, we used other (sometimes even newer references) that did not use this formulation, but we agree that using a non-zero reference temperature has advantages. Accordingly, we have replaced the zero temperature reference with the temporal varying, but volume integrated temperature (see section 2.4). The heat content and surface heat flux are not changed compared to the zero temperature reference, but the horizontal and vertical heat transports are changed by the new reference. In general the main results derived from the heat budget remain valid, but we think that it is still more convincing to use the more reasonable "new" method.  Note that based on later comments in this and other reviews, we have substantially changed the results section concerned with the heat budget. All of these results are now based on the new reference temperature.

3. The description of the results is often ambiguous due to the lack of clear references to the corresponding figures, which may lead to reader confusion. I strongly recommend that the

authors improve the numbering of their figures and carefully review the manuscript to ensure that each figure and its subplots are cited appropriately within the relevant text. Please refer to my comments below for specific suggestions.

We fully agree that at many occasions the references to the figures need to be clearer. In addition to following the suggestions given below, we have carefully checked the entire text and added more references to figures (and subpanels of the figures).

4. Some methodological choices in the study, such as the selection of the 1st and 6th cycle of the simulation, are not clearly justified at the beginning of the paper but are introduced later. To enhance clarity and avoid potential confusion for the reader, I recommend that the authors provide a clear and explicit justification for all methodological decisions early in the manuscript. Please refer to my comments below for specific suggestions.

As suggested, we have added reasons for our choices throughout the manuscript. Please see our detailed response below for specific changes that were applied to the manuscript.

5. The description of the results is often intermingled with elements that belong in the discussion, which detracts from the clarity of the paper. I recommend that the authors more clearly separate these two sections, while also enhancing the structure and clarity of the methods section.

We thank the reviewer for helping us better structure the manuscript. We followed the specific suggestions below and re-edited the results section to achieve a better separation between reporting and interpreting the results. Note however that some choices for the analysis are based on previous results and therefore a certain interpretation is needed to explain these choices.

**Minor – Detailed Comments:**
Line 104: Could you clarify the specific rationale behind selecting the 1st and 6th cycle of the model for analysis? This section is where you should define the key characteristics of the simulations and justify your choices, ensuring that readers understand the reasoning behind them. Providing this information early on will help familiarize readers with the terminology and framework, allowing for a clearer interpretation of the results later in the paper.

We agree that this should be explained already at this point and have added the following statement:
"The first cycle is selected, as it is closest to the observation-based initial state and especially in the context of high-resolution modelling, often simulations only cover the forcing period once. At the same time it is subject to a strong model drift that will be investigated later. The 6th cycle represents an equilibrated model state. Because it had the longest spin-up time, model drift is minimised." (Lines 106-109).

Lines 125 – 130: I recommend that the authors present a more structured and clearly organized overview of the different simulations/experiments conducted, along with their key characteristics. This could be achieved through a matrix or bullet points, explicitly highlighting the distinctions between the two experiments. A clearer presentation of these

details will enhance the readability of the manuscript and provide readers with a solid understanding of the analysis that will follow.

Thank you for this suggestion. Nevertheless, as the manuscript is mainly based on two experiments with the only difference being the initial conditions (and subsequent model drift), we think a table would provide little information. Instead, the explanation of what the cycles represent in the context of our study will also help to better understand our modeling approach.

Line 134: Does that mean that the results will be the same with other baseline approaches (e.g. Smith et al., 2025)?Clarify.

Of course, different baselines result in different MHW statistics, but there are many similar baselines. For example, whether one uses the 1980-2009 or 1991-2020 climatology or any other possible 30-year timespan does not change the qualitative results of this study. We have added this statement to the text (line 139).

Line 172- Equation 2: I understand that this is the conventional approach to formulating horizontal advection flux through each boundary of a given area or section. However, as demonstrated by Lee et al. (2004) and Kim et al. (2006), this formulation renders the conventional temperature flux through a partial boundary problematic. Their research highlights that temperature flux, when expressed in this manner, becomes ambiguous because mass is typically not conserved through a partial interface. Consequently, the advection flux, as derived from the Gauss/divergence theorem, cannot be meaningfully decomposed into western, eastern, southern, and other directional components as presented here. Instead, they propose an alternative formulation that ensures the temperature flux through a partial section remains meaningful by referencing it to the domain-averaged temperature. I advise the writer to carefully read the studies and modify their calculations according to what they would like to represent (total advection flux or advection through a partial boundary)

Thank you for your suggestions to make the heat budget analysis more meaningful. After reading the suggested literature we have replaced the zero temperature reference with a volume integrated reference temperature. As described above, we have made substantial changes to the analysis of the heat budget and its relation to individual MHWs. This new analysis is based on the updated formulation of the heat budget (see section 2.4).

Line 180: Which depths does the integration happen?This needs to be clarified from the beginning to provide the readers with a solid understanding of the analysis that will follow.

We thank the reviewer for pointing out that the integration part was unclear. Actually, this belonged to an earlier manuscript version, where the heat budget was shown in larger depth ranges (multiple model layers). However, we have decided to only show individual model layers for the submitted manuscript, such that there is no vertical integration used in the manuscript. This has been corrected.

Lines 200-202: Which figure do these results correspond to? Please provide a clear reference to the corresponding figures.

We thank the reviewer for recognizing that the correlations were misplaced in the text and have moved them to the correct place with a figure reference.

Line 209: Do you mean Figure 3a-d? The figure subplots should have a different numbering/letter in order to tell them apart more easily.

We have grouped the different depths into a single subplot. To clarify that we have removed the space between the sub-panels and specifically refer to the upper 2 rows of subplot 3a,b. The depths are indicated in the panels.

Lines 210-214: Please, provide a clear reference of the results described here to the corresponding figures.

All these results refer to figure 3, which is referenced in the beginning of the paragraph. We included several distinct references to individual sub-panels.

Lines 214-215: Could you clarify further the connection between the MHW threshold behavior at 2200m and the mid-depth ocean warming observed in the 1980s compared to the 1990s? The relationship between these two aspects is not immediately clear.

We thank the reviewer for spotting this mistake. The fixed (30-year) baseline period used here was 1980-2009 (not 1991-2010 as previously stated). Therefore, excluding the LAST ten years lowers the threshold temperature and therefore leads to even more MHWs after 2010 compared to the fixed baseline (1980-2022).

Line 217: To ensure clarity and consistency for the reader, it is best to use a single term throughout the text. Either "detrending" or "linear increase in baseline" or even better find more intuitive names for the experiments and use throughout the text. This avoids potential confusion

To avoid confusion we have renamed the baselines. We refer to the former "linear baseline" as "detrended baseline" now. Along with this change we always use the term detreding when referring to the detrended baseline.

Lines 218-219: "*Therefore, it is not possible to recover the MHW statistics from non-detrended temperatures in the 6th cycle by detrending the 1st cycle (e.g. 1st-linear does not match 6th-fixed/WMO)*". Could you clarify whether you mean that one should not expect similar MHW statistics when comparing non-detrended temperatures (which include both climate trends and model drift) to temperatures detrended using only the first cycle? What would be the rationale for such an approach? In the first case, strong model drift is present, whereas in the second, detrending removes real-world climate trends. It is unclear why one would expect comparable results under these conditions. Additionally, the distinction between the first and sixth cycle temperatures should be more rigorously defined in the Methods section using precise scientific terminology.

The main point here is that often models are detrended in order to remove the model drift. There is no way to know which part of the trend is forcing and which part is model related.

Essentially, this means one can not use the first cycle with a fixed baseline, because the usual way to deal with model drift also removes the forcing related trends (as one would expect). Ideally one would like to derive the fixed baseline statistics of the 6$^{th}$ cycle, from the 1$^{st}$ (to avoid a multi-centennial spin-up that is very costly). This does not seem possible. We have re-structured the corresponding paragraph. Also, we have clarified the differences in the temperature evolution of the two cycles in more detail in the method section.

Line 222: Did you mean Figure 3a, b? Additionally, the red lines in Figure 3a and 3b appear similar, could you clarify where the dissimilarities are?

The MHW statistics are shown in figure 3,a,b. The next sentence referred to the temperature timeseries itself with an example shown in figure 2. Overall, the red lines are very similar in 3a,b . In particular, at the surface. We have added correlations between the different timeseries here to make this statement clearer. For example, at 1000m depth, the 6$^{th}$cycle shows a weaker increase in MHW days after 2015 compared to the 1$^{st}$ cycle. This is reflected in a correlation that is lower than 1.

Lines 224-225: Problem with the syntax of this sentence.

The sentence was restructured.

Lines 226-227: What do you mean disadvantages due to finite length of the timeseries. Could you elaborate this more so that the reader understands it better?

We have added to the following explanation to the manuscript:
"A moving average leads to a loss of data at the beginning and end of the timeseries, or the averaging window has to be modified at the beginning and towards the end which can introduce spurious signals. Therefore the non-linear shifting baseline is not further discussed in this manuscript." (Lines 2046-248).

Figure 3d: The two regions should be highlighted in different colors and explicitly referenced in the text for clarity. Additionally, why are only the results of the WMO shown, while those of the linear 1st and linear 6th are omitted? Also, it would be helpful to include a brief description of what the 1st and 6th cycle represent in the caption of the figure, for the benefit of the reader.

As for the entire Atlantic, the detrended baseline results show almost no difference between the decades. They would only show 4 additional straight lines close to zero that do not add information. To reduce the amount of lines in the subplot, we have decided to omit them. As colors indicate the baseline (to be consistent with panel 3c), they were not changed. The different regions are indicated by the different line styles (solid and dashed). We added a short description of what the cycles represent in the caption of figure 3.

Line 229: Could you clarify how the MHW statistics were calculated? Were events identified separately at each grid point and depth level across the entire Atlantic, followed by averaging over longitude and latitude? Or was the Atlantic first averaged at each latitude and longitude, reducing the dataset to only the vertical dimension, with the MHW detection algorithm then applied separately at each level? Or were MHWs first identified at the

surface, with MHW days at depth determined based on the timing of surface MHW events? Please clarify the methodology used for constructing the vertical MHW dataset, preferably in methods section.

Additionally, what explains the minimal differences between the 1st linear and 6th linear experiments? Why are the differences in the number of MHW days between the first and last 10 years so small?Clarify. Also what about the differences between the 1st and 6th linear in FIgure 3d?why are they not mentioned anywhere?

The Hobday et al. (2016) definition of MHWs is defined for a single timeseries. We applied this definition to all grid points in the horizontal and vertical individually (i.e. there is no spatial information involved in the detection, but only the local temperature evolution at each grid point). This description can be found in the method section (lines 126-127): "MHWs are defined locally, meaning the definition is applied separately at each individual grid point on the three dimensional grid without considering information from other grid points."
Averages (e.g. for the Atlantic) were calculated afterwards on the MHW statistics themselves. The small difference between the detrended baseline result indicates that the major difference between the decades was a long term warming/cooling (depending on depth) with little differences in the shape of the distribution (standard deviation and skewness). The latter is removed by applying the detrended baseline, such that differences between the decades could only arise if the temperature distribution changed shape. This however is probably less important in most regions, reflected in minor differences between the decades.
As mentioned in our response above, the detrended baseline results are omitted in Figure 3d, because they would double the amount of lines in the figure, without adding any interesting insights. However, to avoid that other readers have the same questions we briefly mention these thoughts in the manuscript (lines 254-257):
"When using a detrended baseline, changes in the mean temperature between the decades are mostly removed and the resulting changes in MHW days are almost zero. This indicates that changes in the shape of the temperature distribution (standard deviation, skewness) that could lead to differences between the decades with a detrended baseline played a minor role."

Lines 236-240: This section leans more toward discussion rather than results, which should focus solely on reporting numbers, percentages, and observed changes. Any further interpretation or commentary should be reserved for the discussion section.

Here we mainly try to motivate the choice of the two regions, by shortly describing the most important dynamical differences and how they lead to the shown results. Although the sentence could go to the discussion we think it is important to briefly mention these differences between the regions when showing the results.

Line 241: "in any case" is not an appropriate scientific expression. Rephrase

The word was replaced.

Lines 242-248: This section leand more towards conclusion than results.

We have deleted the paragraph from the results section and included it in the "Discussion and Conclusion" section.

Lines 246 - 247: What is the rationale behind selecting this specific type of simulation for your analysis? A clear explanation and objective justification for this choice are necessary, either early in the methodology, where all the steps of your methods should be clearly defined, or here.

Although we fully agree that such choices would be ideally all described in the methods section, the choice of the 6$^{th}$ cycle is fully based on the results described in section 3.1. Therefore, we can only justify at this point why we use the 6$^{th}$ cycle in the following analysis. The main reason is that the first cycle can not be used to study long-term changes in MHW statistics below 100 m (this is the main result of section 3.1). However, it is these changes that we aim to study in the following. To keep the manuscript concise we decided to focus on just one experiment (and baseline) to avoid too many sub panels and overly long and complicated descriptions of the figures. We have expanded on our decision in the manuscript (lines 266-272):
"In the following we focus on the characteristics of MHWs detected by applying the fixed baseline in the 6th cycle. As argued above the 1st cycle can only be used to study MHWs relative to a detrended baseline, but we are explicitly interested in studying the impact of long-term changes in the surface forcing and ocean circulation on MHWs. Even though temperature trends in the deep ocean are highly uncertain due to the lack of long-term observations, the well spun-up 6th cycle is regarded as the best estimate available."

Line 252: Why did you select maximum intensity as a metric for comparison instead of mean intensity? A clear and well-supported justification is needed. Also, while the model generally underestimates maximum intensity across most of the Atlantic, it appears to overestimate it in the Gulf Stream region. Therefore, this argument is not exactly correct.

We chose the maximum intensity as it is the most intuitive and easy to interpret metric and just states how much the temperature deviated from the "normal" temperature in the maximum. It is therefore not dependent on the length of the MHW or the exact temperature evolution (e.g. a very strong temperature anomaly followed by a sustained weak temperature anomaly could result in a relatively weak mean intensity). We have also analysed other metrics, such as the mean, or cumulative intensity. The choice of the intensity metrics ultimately depends on the exact scientific question (e.g. impact on marine species). With the physical oceanography view of our study, the choice was to focus on the most extreme values of the temperature distribution (the days with the highest anomalies).
The model indeed shows a slightly stronger intensity of MHWs on the northern GS side. We have mentioned this in the manuscript, although it does not change the general statement that the model underestimates the maximum intensity in most regions.

Line 254: What do these statistics represent?an average over the Atlantic? This should be mentioned in the text. not only in the figure caption.

Yes, it is an average over the Atlantic, which was added to the text.

Lines 263-265: How can model and observations agree on regions with longer and shorter

durations but at the same time the differences between high and low durations be more pronounced in the model? That sentence is confusing. Clarify

We thank the reviewer for pointing out this unclear formulation. We have replaced it with the following sentences (lines 289-291):
"In particular between the equator and 20°N, VIKING20X shows very long, but OISST very short MHWs. In most other regions the model and satellite data agree on whether MHWs are comparably long or short (e.g. long MHWs North of the GS separation and short around the UK), but VIKING20X generally overestimates the duration."

Line 268: How does the vertical resolution of the model play a role when talking about differences in surface MHW characteristics? Unless the writers mean something else. Please clarify.

The vertical resolution can be very important, if shallow mixed layers develop during MHW events. In this case the surface heat flux is mixed over a too large volume in the model and therefore the temperature evolution near-surface is not correctly simulated (neither the model, nor the satellite product represents temperatures directly at the surface). We have clarified this in the manuscript.

Line 270: The term "broken section" is not scientific. Please, rephrase.
Also, it is unclear how the black lines in Figure 5 correspond to the regions mentioned. Are they representing a simple vertical section along specific points across the Atlantic? If so, the exact latitude and longitude coordinates defining the start and end of these sections should be clearly documented in a table rather than solely in the figure maps, ensuring reproducibility. Alternatively, do these lines represent a spatially averaged area around the sections? This distinction needs to be clarified. Furthermore, the rationale behind selecting these specific regions should be explicitly stated. What criteria were used to define these areas, and why were they chosen for analysis? A clear justification is necessary.

We have just replaced it with "section" as it is shown in the figure that the section consists of multiple sub-sections. As the section follows the model grid, the end-point coordinates are not sufficient to reproduce the section. Therefore an additional table would not increase reproducibility in our opinion. The script to how the section was extracted from the model output is uploaded along with the research data. The section shows profiles at individual grid points, not an area average. We have added a short reason for the choice of the section, which is mainly to show dynamically different regions (e.g. subpolar gyre, western boundary current systems, mid-atlantic ridge,...) in a single plot.

Lined 271 - 272: Syntax error in the sentence.

The sentence was corrected.

Line 275: Which area does this sentence refer to?
It refers to regions that are deeper than approximately 4500 m (although an exact depth can not be stated as it is not known where model drift still affects the temperature). This was added to the text.

Lines 277–278: It is unclear whether this statement is directly related to the preceding sentence or introduces a separate result. Additionally, the specific section to which this sentence refers is not clearly defined. Clarification is needed to ensure coherence and precision.

The statement refers to the fact that the sections shown in figure 3a,b have mostly opposing patterns, i.e. a long  mean duration of MHWs coincides with a low frequency and vice versa. The figure reference was added to the text and the sentence was clarified.

Line 290: does Labrador Sea correspond to Canada section in FIgure 5? Not clear. It is better to keep a consistent name of your areas both in the text and in the figures.

Yes, the Labrador Sea is seen as the section between Greenland and Canada.  This reference was added to the text.

Line 295: For clarity and to benefit the reader, it is essential to use the same regional names in the text as those used in the figure. Additionally, please refer to the corresponding subplot in Figure 5 to further enhance clarity.

The figure labels only state the name for the land features to give a geographical context. There are too many different currents and ocean regions to label all of them in the figure, therefore we always included a reference to the corresponding land masses in the text, or the newly introduced labels for the section vertices (see also our response to your comment on figure 5 below).

Line 300-302: It is important to reference specific subplots of Figure 5 and use consistent terminology for the regions throughout the text. Additionally, the explanation provided here appears to be more suited for the discussion section. Please ensure that the results section remains focused strictly on presenting the findings, without further interpretation.Also the MLD does not increase. Instead it gets shallower.

We have added references to the specific subplots here. We also thank the author for pointing out that we meant to write the MLD decreases. Although we often followed your advice to move parts to the discussion section, we think that in this case a short interpretation of the results is appropriate for the reader to immediately understand the results.

Figure 5: The subplots of this figure need to be named separately for clarity purposes and ease of reading in the text.

Figure 5 does only consist of 5 individual subplots. The vertical lines just mark the individual section segments, but the plot shows a continuous section through the Atlantic. For a better orientation along the section, we have labeled the vertices of the section in figure 5 and 4 with letters.

Lines 212-213: Can you provide evidence to support this, or is this a claim based on theoretical assumptions? It is important to substantiate this statement with relevant data, references, or a clear rationale to strengthen the argument.

Although it states lines 212-213, we assume this comment refers to lines 312-313, based on the order of the comments. This statement is based on additional analysis that is not shown in the manuscript. We have also detected bottom MHWs with the detrended baseline and we found that differences between the baselines are overall small inthe 6[th] cycle, except for the abyssal ocean areas. It is not possible to know whether this is caused by model drift, or by forced circulation changes. Based on the long adjustment timescale of the deep ocean model drift can not be ruled out. We have clarified this in the manuscript.

Lines 316,320,323: You need to reference the relevant figure here, as it is unclear which one the reader should be looking at. Including a specific figure reference will enhance clarity and guide the reader effectively.

We have added more references to the figure subpanels in the respective paragraphs.

Lines 317-318: I do not understand this sentence. Do you mean that the interior and near coastal regions of the ocean, have similar sea-floor, depths? This sounds strange. Also, what does it mean" *even if the sea floor is located in similar depths*"? What would happen if the sea-floor was located in different depths (which it does not, at least comparing the coastal and interior areas of the ocean).

This sentence was just meant to state that the lower duration along the continental slope can not just be explained by the bottom being located at shallower depth compared to other regions of the ocean. For example along most of the western boundary the duration is lower than along the Mid-Atlantic Ridge, but the sea-floor along the Mid-Atlantic Ridge is shallower than along the lower continental slope along the western boundary. We added an additional explanation to the text.

Line 326: It is unclear what is meant by "linear trend" in this context. Does it refer to a trend in temperatures or a trend applied to identified MHWs? Please clarify the specific variable or process being referenced by the term "linear trend."

This refers to a linear trend in the MHW characteristics themselves, not a trend in the temperature for example. This was clarified in the manuscript.

Lines 327-329: These sentences require improved grammatical flow and syntax, in addition to clear references to the appropriate subplots of Figure 6 for enhanced clarity. What specific trends are being referred to? Are these linear trends in temperature, or do they pertain to trends in MHW duration or intensity? The current phrasing is unclear and needs further specification.

We have added a more clear description of the trends that are described here. Also, we have simplified the sentences for better readability. As the paragraph is rather short and figure 6 is explicitly referenced in the beginning of the paragraph, we did not introduce additional references.

Lines 339 - 340: Please provide a justification for your choice of the WMO baseline over the linear one. Additionally, clarify where you demonstrate that the conclusions are independent of the baseline selection.

We have chosen to focus on the fixed baseline here, since it allows us to investigate the impact of long-term trends. As shown in the previous part of the manuscript, a long model spin-up is needed to do so. Having such a long model spin-up at 1/20° resolution is very rare and therefore including the trends provides unique insights that are new and not easily obtained.
The following text was added to the manuscript (lines 267-270):
"As argued above, the 1st cycle can only be used to study MHWs defined with detrended baseline, but we are explicitly interested in studying the impact of long-term changes in the surface forcing and ocean circulation on MHWs. Because including the trend requires a very long-model spin-up that is rare at 1/20° resolution, this provides unique and novel insights into the characteristics of MHWs and the impact of long-term temperature changes."

Lines 348-349: The statement regarding the mean frequency being similar, yet higher in VIKING20X, is contradictory. Please clarify how these two observations can be reconciled.

We have clarified in the manuscript that the similarity only refers to the temporal variability (reflected by a high correlation of 0.94), but there is a difference in the time mean MHW frequency.

Lines 350-351: These sentences belong to discussion

Although we agree this is an interpretation of the results, it is also the justification to use VIKING20X over ORCA025 for the majority of our analysis. In particular, it strongly contributes to considering the 6th cycle of VIKING20X as our best estimate available. Therefore, we think that a short statement which dataset we consider more realistic is needed at this point.

Lines 355-357: In addition to the fact that this sentence belongs in the discussion section, how can the entire Northeast Atlantic MHW features be attributed solely to the Mediterranean outflow, rather than interactions between the North Atlantic Current and waters of (sub)polar origin? Please provide evidence to support this claim.

We have moved parts of the paragraph to the discussion section, but a certain explanation of the results, mainly linking changes in the MHW characteristics (figure 7) to the differences in the temperature and circulation of the model configuration (figure A1), belong to the results. We have added more figure references here to indicate that these sentences are a description (and interpretation) of the result figures.
We mainly argue that the high vertical temperature gradient between the Mediterranean Outflow Water (MOW) and North Atlantic Deep Water (NADW) or Subarctic Intermediate Water (SAIW) is responsible for strong MHWs (see Liu & Tanhua 2021 for a discussion of water masses in the region). Individual MHWs are not necessarily driven by changes in the MOW properties itself (or changes in the other water masses involved). They are more likely caused by vertical or horizontal advection anomalies in the presence of the strong gradients, or internal waves that displace the isotherms. Figure 1 below clearly shows that the intensity of MHWs is strong where the isotherms are closer together, representing a strong gradient. The transition between MOW and NADW between 30 and 50°N is mostly associated with a change of temperature in vertical direction, while further north the transition between MOW

and SAIW also has a horizontal component. South of 30°N the vertical gradient is smaller (distance between isotherms increases) and the strongest gradient is shallower than 1000 m. This results in less intense MHW, in particular at 1000 m depth.
We have expanded our explanation in the revised manuscript to clarify the role of these water mass transitions in more detail.

[Figure]

**Figure 1:** 1980-2023 mean MHW maximum intensity (a) and mean temperature (b) along a section at 20°W. The labeling of water masses is based on Liu & Tanhua (2021).

Lines 358-370: These sentences belong to the discussion.

See our response to the comment above, as it applies to the entire paragraph. Note that we have also moved lines 377-379 of the initial manuscript to the discussion.

Lines 376: Have you shown that somewhere?supplementary material perhaps?not clear.

We have looked at similar plots as Figure 7 for other depth levels as well. To avoid having a huge number of figures we decided to only show 1000 m as an example here. Nevertheless, we think it is important to mention that the results obtained for 1000 m depth are valid for a larger depth range, although this is not explicitly shown here. We have added this explanation to the text.

Line 391: The phrase "are not in phase" is not the most appropriate syntax here. An alternative phrasing could be: "The surface MHW characteristics differ from those observed at depth." Additionally, you need to specify which subplot of Figure 9 you are referring to, as it is currently unclear where the reader should focus.

It is not necessarily about the mean characteristics, but we want to emphasise that their variability has a different timing at different depths. For example a period of short MHWs in the early 2010s at the surface goes along with a period of long MHWs below 100m. As this is seen in all characteristics, we have added a reference to figure 9a-c. This example was also added to the manuscript. Also, we have replaced "in-phase" with "coherent" variability.

Line 392: Could you clarify what you mean by "explained variance"? Was a statistical test conducted to analyze the MHW characteristics?

The explained variance is defined as the squared correlation coefficient ($R^2$) and chosen here because it is often used in the MHW literature. This was added to the manuscript with an extended explanation of what the metric shows.

Lines 393-394: How do the atmospheric heat fluxes relate to this figure, which focuses on MHW characteristics?

We agree with the reviewer that this reference to the surface heat flux was unclear. We refer to the heat flux here, because it accounts for most of the variability in the surface layer. We have clarified this in the revised manuscript. The sentences now read (lines 433-435): "This means that variability of the surface metrics, which are strongly linked to variability of the surface heat flux as will be shown later, accounts for only a small fraction of the variance in the MHW characteristics at depth."

Lines 396-397: The decoupling of surface and deeper MHWs is generally not attributed to long-term trends. Instead, the distinct variability and decoupling of MHW characteristics are typically the result of differing oceanic processes at the surface (high-frequency variability) and at depth (low-frequency variability). Therefore, the physical relevance of this sentence is unclear.

This is true and the main point we try to make here. Nevertheless, in the presence of strong temperature trends, the long-term warming will cause a coherent increase in MHW statistics regardless of other drivers and could decouple surface from sub-surface variability. For example a long-term warming (that is stronger near the surface) will lead to an over proportional occurrence of MHWs in the upper layers compared to deeper layers, even though the drivers of individual events (e.g. anomalous solar influx or eddy transport) act coherent over the depth layer. We have rephrased the sentence and explained in more detail why different trends could be important. We think this is an information that is worth mentioning even though most people may correctly assume that oceanic processes are more important here.

Line 400: Again, what is explained variance here? Was there a statistical test performed here?

We have added a definition to the manuscript text where explained variance is mentioned first (see our response above).

Lines 403-404: If this refers to a different dataset or data handling process, it is essential to provide a clear reference to the methods section for better context and understanding. Please ensure that the relevant details are adequately explained and linked to the methodology for clarity.

If this comment refers to the scaled maximum intensity, it is just another "standard" characteristic of MHWs, similar to frequency, or duration. This sentence is just a short reminder what the difference between the maximum and scaled maximum intensity is.

Line 406: Please ensure that the subplots in Figure 5 are clearly labeled and referenced in the text. This will help guide the reader to the correct location for the relevant information. The current lack of clarity makes it difficult to understand which specific subplot is being referred to.

This line referred to Figure 5c. We have added this reference to the text.

Lines 406-407: The typical range of MHW variability is unclear in this context. Please ensure that you reference the appropriate figure or subplot to guide the reader. Without this reference, it is difficult to determine which data you are referring to.

Thanks for pointing out this unclear formulation. This refers to the temperature anomaly associated with MHWs compared to the typical range of temperature variability. This explanation was added to the text (line 450):
"This means relative to the typical range of temperature variability, the anomaly associated with MHWs is similarly strong at all depths."

Line 411: The statement that MHWs show no considerable trends at this depth appears contradictory. From Figures 9a-c, it is evident that there is a higher number of events at the beginning of the timeseries, with a decreasing trend toward the end. Please clarify this inconsistency.

We agree with you that there is still a trend towards less MHWs, but compared to all other depth layers the trend is much weaker and the MHW statistics show more variability on short timescales. We have changed the respective sentence accordingly (lines 455-456):
"Between 3000 and 4000 m there is a small (compared to other depth layers) decreasing trend in frequency and duration, but variability on interannual timescales dominates the timeseries"

Lines 412-413: These sentences belongs to methods. Also the phrase "MHWs vary in phase" everywhere in the text. This sentence is unclear and contains syntax errors. It is important to specify whether this refers to the relationship between surface and subsurface layers or if it applies to other layers. Please clarify the intended meaning, or consider rephrasing with a more precise expression for better clarity.

In order to avoid any misunderstandings, we have replaced "in-phase" by either "coherent variability", or "same temporal evolution" which all refer to MHWs in different depths showing increases/decreases at the same time.
Although we fully agree that this refers to the method, this sentence was not meant to introduce a new method, but to remind the reader that MHWs were detected at individual grid points (and depth levels) and thus it is not granted that there is any coherence across the vertical extent of the water column. We have explicitly phrased this as a reminder and included a reference to the method section to avoid confusion.

Figure 9: Depth ranges between Figures a-c is up to 3800m while in Figures d-f up to 1500m. This inconsistency should be addressed. Consistency across the figures is essential for clarity and accuracy in presenting the data.Also in the caption of this figure, a new

simulation is mentioned "*WMO baseline but with the trend removed after calculating the annual mean MHW characteristics* ". However, this simulation/experiment has not been mentioned previously in the Methods section, making it the first reference to it in the figure caption. It would be more appropriate to introduce and explain this simulation in the Methods section to adequately prepare the reader before mentioning it in the figure caption.

The reason for showing only the top 1500 m is twofold. First, the figure focuses on whether the characteristics of deep MHW vary coherently with those at the surface. Therefore, the plot should emphasize the surface ocean. As the variability is clearly distinct between the surface and sub-surface already between 0 and 100 m there is no physical process that could explain a higher correlation between the surface and depth beyond 1500m. Second, the very computationally demanding detection of MHWs throughout the entire ocean (here beyond 1500m) was only done for the fixed 30-year baseline, but not the detrended baseline. Therefore, we can not extend the plot to 4000m

The additional dashed lines do not represent a new experiment, or different method to define MHWs. After applying the 30-year fixed baseline  to the 6[th] cycle of VIKING20X (solid lines) we have removed the linear trend from the MHW characteristics themselves (dashed lines). This is meant to support our statement that different long-term trends at different depth can not explain the decoupling of near-surface and sub-surface MHWs (see our response to the comment on lines 396). We have clarified this in the figure caption.

Line 420: It is not clear which specific subplot of Figure 10 is being referenced here, nor which experiment these results pertain to. It is essential to clearly specify the relevant subplot and provide context regarding the experiment to avoid confusion for the reader.

We agree that it was not always clear to which experiment our results referred. We have added more reference to the specific subplots in figure 10.

Line 422: This sentence is unclear. What is meant by stating that the linear trend had a minor impact on the Cape Verde Archipelago? The linear trend can only affect the MHW characteristics in the Cape Verde region, not the region itself. Please clarify.

We have clarified that this refers to MHWs in the Cape Verde Archipelago region.

Line 423: The MHW coverage differs between the two simulations (Fig. 10a-b), particularly at depth, and is not similar at the surface.Clarify.

Although there are some differences between figure 10a and b in the mixed layer the differences are much smaller compared to other depths. Note that we have replaced the MHW coverage in figure 10 with individual MHWs detected in the (horizontal) mean temperature timeseries of the Cape Verde archipelago region. Nevertheless, the argument is still similar and was clarified in the revised manuscript.

Lines 425-426: I do not understand how can you physically connect temperature anomalies (that are above the 90th percentile) and that have been identified at each layer separately. The application of the Hobday et al., 2016 at each layer does not guarantee spatial coherence of events at surface or at depth and so the events are not demonstrated to be connected from one layer to the other just because there is a temporal continuation of

temperature anomalies. How can you know if the process that is responsible for the development of temperature anomaly at let's say 600m is physically and mechanistically related to the temperature anomaly identified at surface? This is an implicit assumption which can be made only by looking at the figures but it is not physically proven. Especially because these are anomalies above the 90th percentile at each layer. If an anomaly at a selected depth is below the90th percentile for some days but then comes back above the threshold, will it still be considered as a MHW?This is a problem inherent to the statistical MHW framework. Therefore assumptions of the vertical continuation of MHWs should not be made lightly.

We fully agree that the coherence of MHWs in different layers does not allow for any statement about the physical drivers. Following the usual procedure, the temperature anomaly is allowed to fall below the $90^{th}$ percentile for 2 days during a single MHW. If it stays below the $90^{th}$ percentile for longer and then exceeds it again, this is considered a separate event.
To study whether the apparent coherent occurrence of MHWs is related to specific drivers, we have replaced the former heat budget analysis with an event based view of MHWs. For that we have detected MHWs in the spatially averaged temperature of the Cape Verde archipelago. In contrast to the previously used area covered by MHWs, this allows for well defined start and end dates of events. With that we define the contribution of the different heat budget terms to the development of heat content anomalies associated with MHWs. Note that heat, as defined in our study, is linearly related to temperature and therefore a MHW is also a heat extreme. Thereby, it is possible to attribute the heat content change (i.e. temperature increase) related to a specific MHW event to a specific driver. Our results show that indeed, the layers in which MHWs occur at the same time are related to specific drivers or changes in the vertical structure of these drivers.  For example throughout the mixed layer it is clearly the surface heat flux that contributes most. Below the mixed layer lateral and vertical heat transport both contribute to events, while below vertical heat transport dominate the onset phase of MHWs. This is shown in the updated figures 10 and 11 and the corresponding text in section 3.4.2 and 3.4.3.

Line 430-434: You need to reference the appropriate subplot of the figure for clarity. It is currently unclear which part of the figure these sentences refer to.

We have added additional references to the subplots in the paragraph. Please note that this paragraph has substantially changed due to our new approach to investigate the drivers of individual MHW events.

Lines 437: The specific figure being referenced here is unclear, and the concept of "disconnection between the two depth layers" is not well-explained.

We have added more figure references and replaced the mentioned part of the sentence. It now reads (line 490):
"Using the detrended baseline, MHWs additionally occur after 2010 (figure 10b)"

Lines 438- 439: Please provide a clearer explanation of this result, and ensure the appropriate figure is cited for better clarity.

This sentence summarises the previous paragraph and with the added references in the previous two sentences it is now clearer to which figure this is linked. Also, we have added additional explanation how this statement is derived from the comparison of the detrended and fixed baseline results.

Line 441-442: Has this long-term cooling trend seen anywhere?An appropriate figure (subplot) should be cited here.Otherwise this looks as a general statement without proof.

We have added more figure references in the respective paragraph. Although figure 10 has changed, the cooling trend itself is still visible in figure 10a as a continuous decrease in heat content.

Lines 443-444: The similarity being discussed is between the simulations/experiments that identify MHWs using the WMO and linear baselines, not between the baselines themselves. This distinction needs to be clearly defined throughout the text. As it stands, the sentence is unclear and requires better clarification.

Similar variability can be seen in the same model experiment (VIKING20X-6th), but with different baselines (fixed, detrended) used to detect MHWs in these experiments. The first cycle of VIKING20X is not used after section 3.1. This is mentioned at the end of section 3.1, but we have additionally added this statement to the methods section.

Lines 446- 455: This paragraph belongs to the discussion.

This part has been deleted following the updated heat budget analysis.

Lines 457-458: The phrase "hint on connection to the surface" is unclear. To improve clarity, the authors should specify what indicators or evidence suggest a connection between surface and subsurface events

This part has been deleted following the updated heat budget analysis.

Line 464: The base of the mixed layer should be clearly defined in the text or figure legend, as it is not evident from the figures. Additionally, it is essential to reference the specific figure or subplot to ensure the reader knows exactly where to look for the relevant information. For clarity, the layer's depth or definition should be described in relation to the figure presented.

This part has been deleted following the updated heat budget analysis. The term base of the mixed layer is not used any longer.

Line 465: As previously mentioned, the MLHB format utilized in this study introduces ambiguity when representing temperature flux across a partial interface (x, y, z). It currently reflects the redistribution of heat within the domain rather than the change in heat content at the interface due to temperature variations. See my comment above

We thank you for this comment. As outlined above we have changed the heat budget as suggested.

Line 468-472: This description probably refers to Figure 1? however, as no specific figure is cited, it is unclear where these results correspond to. Please clarify by referencing the appropriate figure (subplot) to ensure the reader knows where to look. Currently, this sentence is vague.

This part has been deleted following the updated heat budget analysis.

Line 474 - 475: This sentence is unclear. The sentence cites Figure 11b, discussing the impact of removing the linear trend, while the caption of Figure 11b describes the ocean heat content change based on the linear baseline. Please clarify the relationship between the two, as the current wording is contradictory.

This part has been deleted following the updated heat budget analysis. The new figure 10 now shows only heat content anomalies, not the heat content changes.

Lines 475-476: Have you shown or read that before?Otherwise it is an arbitrary statement.

This part has been deleted following the updated heat budget analysis.

Lines 477-478: This sentence is confusing. Please clarify or rephrase.

This part has been deleted following the updated heat budget analysis.

Lines 478 - 483: Once again, the specific location of these results is unclear, and it is not evident which figures to refer to. Additionally, why is the description of the heat content provided for only one experiment/cycle used in the study, and not for both? Is there a notable difference between them? Please clarify.

This part has been deleted following the updated heat budget analysis, but the reason for using only one cycle here is that the heat budget (and other analysis presented here) is very extensive. We have focused solely on the 6$^{th}$ cycle of VIKING20X, after discussing the impact of model drift. We think the well spun-up 6$^{th}$ cycle provides a more realistic temperature evolution (at depth) compared to the 1st cycle.  Trends in MHW characteristics derived from the 1st cycle would be very different, but variability on shorter scales is expected to be more similar.

Line 484: This statement requires proper citation of figures. Currently, it is unclear which figure is being referenced. Please ensure that the relevant figures are cited explicitly for clarity.

This part has been deleted following the updated heat budget analysis.

Lines 486-487: Could you please clarify the depth of the mixed layer base? Additionally, the colors used in Figure 11e are not distinct enough to clearly distinguish the different processes described. I recommend adjusting the color scheme for better clarity and differentiation of the processes.

The term mixed layer base is not used any longer in the revised manuscript.

Line 497: Could you clarify what you mean by "changing relative importance"? The expression is unclear. A rephrasing or further explanation would help ensure clarity.

This part has been deleted following the updated heat budget analysis.

Lines 501-511: This paragraph is more conclusion section and not results.

We agree that the entire paragraph belongs to the conclusion and was therefore removed. Parts of the content were added to the "Summary and Conclusion" section.

Figure 11: Could you specify the climatology period used to calculate the ocean heat content anomaly plot? This is currently unclear.

We have applied the same methodology as for the heat waves. As we have updated the figures corresponding to the heat budget analysis, the new figure 10 shows the heat content anomaly based on the fixed 30-year (figure 10a) and detrended (figure 10b) baselines. This information was added to the caption.

Line 546: Could you clarify what you mean by stating that the WMO and fixed baselines are not applicable below 100m? What specific application or context are you referring to?

We have clarified our statements in the revised manuscript and removed the word "applicable". The main argument is that in the presence of model drift, MHWs detected with a fixed baseline are only reflecting the model's adjustment to the initial conditions. Therefore, the detected MHWs are not reflecting any processes that occur in the real ocean, such as circulation changes forced by wind, heat- or freshwater flux variability. In that sense the model drift dominated MHW statistics are not useful for any kind of application.

Line 565: Could you clarify what you mean by "in the upper ocean, temperatures cover a large range"? Temperature is a fundamental and ubiquitous property of the ocean, so it would be helpful to specify what aspect of the temperature variability or range you are referring to.

We thank you for pointing out this unclear formulation. We were referring to the range of variability, but replaced this formulation with the more simple statement that the upper ocean temperatures are much more variable than deep ocean temperatures (higher variance).

Lines 585-586: Once again, I do not believe the influence of the Mediterranean Sea Outflow (MSO) has been adequately demonstrated here. It appears to be inferred based on unclear deductions. Please provide compelling evidence to support the claim that the variability described is indeed related to the MSO, or alternatively, clearly state it as a speculative observation. Additionally, which specific depth are you referring to? Given that the study examines multiple depths throughout the water column, this statement may not hold true across all depths analyzed. Please be more precise in your reference to the depth in question.

We have extended and clarified the link between MHWs in 1000 m depth and the mediterranean outflow. Please see our response to your comment (Lines 355-357) above and figure 1 of this response letter. We have also added a specific depth range (0-1500 m) we are referring to. This is the depth range where the transition between water masses is often associated with large temperature gradients.

Line 588 - 591: For which of the experiments conducted here does this apply? Or are these observations valid for both experiments? Please clarify.

These observations are indeed valid for both experiments, which was added to the text.

Line 593: A decrease in extreme positive temperatures? How is this observed? Which figures illustrate this? Please clarify.

This was just used as a synonym for MHWs, but we agree that it may be confusing and decided to stick with the term MHW.

Lines 614-615: Which publication. give examples.

We have added a few references here that are of particular relevance as examples.

Line 619: The word "pitfall" is not an appropriate word for a scientific paper. Use, limitation or challenge instead.

We have replaced the word as suggested.

---

## Author Comment (AC3)

**Response to reviewers**

**Review CC2**

Thank you for your evaluation of our manuscript and the detailed questions on the model strategy and simulated processes. Please find a detailed response to your comments below.

**1 Modelling**

Forcing conditions are applied to the model and presumably include heat, currents and air circulation, which are all mentioned in the study without explicitly reporting the full scope of said conditions.

Simulations are said to be performed using the VIKING20X following the OMIP-II protocol, prescribing six consecutive simulations spanning the 1958 to 2019 time-frame, with the first one initialising from WOA13 data and oceanic conditions at rest, while each of the following cycles are initialised using the final oceanic conditions of the previous cycle; each cycle is then extended up to 2023 to analyse the 1980 to 2022 time-frame. In particular, only the first and sixth cycle of each series are analysed: I infer this is done to observe an immediate response to the forcing conditions in the first cycle and the influence of model drift and model spin-up in the last cycle. While I believe this is needed due to the limited timeframe in which data is available I could not find evidence of the validity of this cycle-based approach for the simulations either in this paper or in the provided reference (Tsujino et al., 2020), as after a new cycle has started oceanic conditions at a certain time t in the time-frame provided (1958-2019) would instead be mapped in the simulation to a time $t' = t + (n − 1)\Delta$, where n is the number of the cycle being computed and $\Delta$ is the length of the time-frame, to my understanding.

In general, all cycles allow us to infer the response to the surface forcing, as they are the same in all cycles. The surface forcing itself is a state-of-the-art dataset used by numerous modeling groups to study the past evolution of the ocean, which is described in *Tsujino et al. (2018)*.
The first cycle is here used to decipher the role of model drift. Although this is a simplification, the temporal evolution of the first cycle contains forced variability (related to the surface forcing), intrinsic variability (related to stochastic processes) and an adjustment to the initial conditions (model drift). Cycling through the atmospheric forcing multiple times allows for the model to dynamically adjust, reducing the drift after initialization. Each cycle has different initial conditions that are closer to the model's response to the applied surface forcing. Therefore, in an ideal case, the temperature evolution of the 6th cycle would only reflect forced and intrinsic variability. In reality the deep ocean may have not reached this equilibrium yet. Nevertheless, in the 6th cycle the impact of model drift is clearly reduced compared to the 1st cycle. Of course there are other spin-up strategies, but the goal is always to get closer to the model's equilibrium state, such that any spurious trends are minimized. The strategy we follow here is suggested by the OMIP-II protocol and indeed described and justified in Tsujino et al. 2020. The need for multiple cycles is for example explained by Tsujino et al. 2020: "However, in preliminary JRA55-do-forced (OMIP-2) runs conducted by many modeling groups, decline and recovery of the Atlantic meridional

overturning circulation (AMOC) occurred during the first few cycles before it reached a quasi-steady state."

**2 Results**

The influence of geothermal activity on MHWs is not mentioned in the paper, and as such I would like to inquire if it is speculated to be noticeable, especially on bottom MHWs, or would stable geothermal activity not impact MHW formation due to their statistical definition?

This is an interesting question. Stable geothermal activity would not lead to a MHW as you point out. Locally the temperature would be warmer than in the surrounding, but constant in time. Nevertheless geothermal activity is very unlikely to be constant and could be related to MHWs in very active regions, e.g. around the Mid-Atlantic Ridge. However, it would still be a very local process that is way too small to be simulated in the model with a resolution of 3-5 kms. Additionally, the temporal variability of this heat flux in specific locations is not known and therefore it is not possible to test its impact in our model configuration.

The effects of model drift are shown to be greatly reduced in the linearly-increasing baseline. Being model drift defined as the adjustment of the simulated environment to unknown initial conditions, could it be argued that the primary effect of these unknown initial conditions is the temperature rise, thus reducing the model spin-up time needed, and that other lesser effects of said conditions are higher-order corrections?

If we understand you correctly, the question is whether a linear baseline approach reduces the spin-up time. Indeed the main effect of the initial conditions is to introduce a long-term adjustment that is mostly linear. There are also some non-linear effects, as evident in figure 3 for example. Nevertheless, these effects are smaller and can often be neglected. Thus applying the linear baseline in a model experiment with a shorter spin-up is possible (at least in our model) without introducing major errors (when the $6^{th}$ cycle is regarded as our best estimate).

The simulations performed with this new protocol are said to not be decidedly more realistic than previous ones. Since the main difference from previous models is the impact of mesoscale dynamics, which could either have a cumulative impact or be averaged out over larger portions of the ocean, could these lower scale dynamics be seen as higher-order terms in the model approximations? If so a convergence interval should be defined, where the model can be argued to be more realistic, while outside of it higher-order corrections may not yield better approximations.

The model experiments are more realistic than previous ones used in many aspects. In particular this is related to the resolution of the model, which for example allows for a much more realistic path of the Gulf Stream and North Atlantic Current compared to coarser resolution models. This is one aspect that does not average out and can not be adequately represented with an eddy parameterization in a coarser resolution model. A correct GS/NAC path is very important locally for the characteristics of MHWs. Additionally, individual mesoscale eddies can generate MHWs (see for example Großelindemann et al. 2022). Such MHWs would be missing if an eddy parameterization is used in a coarser model, which can introduce the net effect of eddies on tracer gradients, but not the effect of individual eddies.

To define a proper convergence interval where the MHW statistics are no longer sensitive to the grid resolution, one would need the same experiment at various resolutions. However, we only have two resolutions (0.25° and 0.05°) available. In any case, when looking at extremes, non-linear processes are likely important and as shown in our study there are major differences between the two resolutions. This clearly indicates that coarser resolution models do not capture the effects of mesoscale variability on MHWs.

Being MHWs defined as events lasting at least five days, and since MHWs divided by less than two days are considered the same MHW, would this merging of MHWs cause problems for the MHW frequency data in areas where they tend to have longer durations along the length of the simulations?

As you point out, the duration and frequency are often directly related in particular as MHWs become very long. A MHW that lasts 365 days can, by definition, only occur once a year. Allowing for a gap of 2 days therefore leads to longer MHWs, but less MHWs. This however, is not necessarily a problem. An ecosystem may not recover from high thermal stress within 2 days, thus it is more reasonable to merge events. Using 2 days instead of any other gap is of course a choice and will often not be exactly the time an ecosystem needs to recover (and to allow the assumption that two events are independent). As our aim here is to provide an Atlantic wide three dimensional dataset as a complement to already available datasets for the surface, we choose to apply the common definition allowing for a 2 day gap.

The heat budget present in this study doesn't contain, at least from what is shown, the influence of the night-day cycle. Since MHWs at shallow depths are shown to be highly responsive to external conditions it could be an interesting forcing condition, but it may very well be averaged out over the multi-decadal time-scales used for the simulations, and it would outgrow the focus of the study on greater depth MHWs.

The day-night cycle is included in the model forcing (for example in the shortwave radiation flux). However, following the well-established definition of Hobday et al. (2016) the analysis is performed on daily means of the temperature. This is to maintain comparability to numerous other publications using the same approach. The most important motivation for using daily data is that temperature extremes are considered to have more severe impacts if they are sustained for sufficiently long time. Variations in the day-night cycle might be relevant for some species as well, but major ecosystem damages are expected when temperatures are higher than usual for at least several days.

---

## Referee Report (RR1)

I congratulate the authors for the substantial changes they have performed in the manuscript. I believe the updated version is more coherent and especially the discussion section is more clear than before. Therefore I recommend publication, after a few very minor comments below are addressed

**Minor comments:**

Lines 110-113: They better moved after line 99 because there meaning is closer to that paragraph. Lines 191: Is there a reason why the fixed43yr was not used here for the calculations of ocean heat content anomalies? If so, this has to be justified somewhere. Also it has not been used anywhere in the anlays after that (e.g. in the Cape Verde analysis) and no justification was provided for this. Line 624: "These MHWs are not be necessarily caused by.."

---

## Author Response (AR2)

**Response to Reviewers**

**Report #1**

Thank you for your thoughtful addressing of the issues raised in my initial review. I have listed below only a few minor points to be clarified:

Thank you very much for helpful suggestions and for re-evaluating our revised manuscript.

Line 446: This is normalisation is commonly called the continuous severity index, or 'severity' for short (established by Sen Gupta et al. 2020 – see their methods section, an extension of the discrete severity categories from Hobday et al 2018), I suggest you update your paper to use this terminology to avoid confusion with other studies.

Thanks for your suggestion. We have updated the manuscript to use maximum severity instead of scaled maximum intensity (in contrast to Sen Gupta et al. we do not calculate the continuous index, but only its maximum).

Line 431 - correct to "in the 2010s"

Corrected

Line 468-469: I find this a little cryptic – 'more similar' to what? Can you rephrase this sensence. We refer to the similarity between MHWs detected with the fixed and detrended baselines close to the surface. This was added to the text.

Line 501: Change "extend" to "extent" Corrected

Line 504-505: Is it not more correct to say here that the mixed layer MHW's are driven by the residual terms, of which the air-sea heatflux is likely to be the largest (as far as I can see, you don't look at the relative size of the terms in the residual.

Yes this is true. We did not look at the vertical and horizontal contributions to the residual term. Therefore, we have changed the sentence according to your suggestion.

We are still very confident that the vertical mixing is much more important than the lateral diffusion across the lateral boundaries and dominates the residual term. This is because for a single depth level the area of the domain is much larger than the cross-sectional area of the lateral boundary. Also, the vertical temperature gradient is much larger than the horizontal temperature gradient. Note that the air-sea heat flux affects only the first model level directly. Heat gained in the surface model layer is mixed downward, such that it is part of the residual term in all levels below.

**The sentences now read:**

"Within the mixed layer, MHWs detected with the detrended baseline are almost exclusively driven by the air-sea heat flux (uppermost vertical level) and the residual term. Note that the air-sea heat flux as defined here only contributes to the budget of the uppermost model level. Downward mixing of this heat to deeper levels is part of the residual term and likely the dominant contribution throughout the mixed layer."

**Line 527: I'm struggling to see a minimum at 1300 m, please clarify?**

Indeed, the minimum is not at 1300 m, but the vertical temperature gradient becomes small compared to the upper water levels as it declines towards a minimum in approximately 1600 m depth. We have clarified this in the manuscript.

**References:**

Hobday, A. J., Oliver, E. C., Gupta, A. S., Benthuysen, J. A., Burrows, M. T., Donat, M. G., ... & Smale, D. A. (2018). Categorizing and naming marine heatwaves. Oceanography, 31(2), 162-173.

Sen Gupta, A., Thomsen, M., Benthuysen, J. A., Hobday, A. J., Oliver, E., Alexander, L. V., ... & Smale, D. A. (2020). Drivers and impacts of the most extreme marine heatwave events. Scientific reports, 10(1), 19359.

**Report #2**

I congratulate the authors for the substantial changes they have performed in the manuscript. I believe the updated version is more coherent and especially the discussion section is more clear than before. Therefore I recommend publication, after a few very minor comments below are addressed

Thank you very much for your detailed comments on our first manuscript version and for re-evaluating the revised manuscript.

**Minor comments:**

Lines 110-113: They better moved after line 99 because there meaning is closer to that paragraph.

We have moved the paragraph to line 99.

Lines 191: Is there a reason why the fixed43yr was not used here for the calculations of ocean heat content anomalies? If so, this has to be justified somewhere. Also it has not been used anywhere in the anlaysis after that (e.g. in the Cape Verde analysis) and no justification was provided for this.

We have not used the fixed43yr baseline to reduce the number of subplots, to save computing time and because it does not provide much additional information. The 43yr baseline was only introduced to study the impact of including more years when calculating the MHW threshold. As the differences between 30yr and 43yr fixed baselines are overall small and using 30 years is the more common choice, we have only used the fixed30yr baseline after the comparison in section 3.1. We have added this statement to line 191.

Line 624: "These MHWs are not be necessarily caused by.."

This sentence was corrected.

In addition to the changes mentioned in our response above, we have applied a few minor changes, correcting typos or improving grammar. All changes are marked in the tracked changes pdf document.